# Learning Latent Action World Models in the Wild

**Quentin Garrido** [1]   **Tushar Nagarajan** [1]   **Basile Terver** [1,2]   **Nicolas Ballas** [1]   **Yann LeCun** [3]   **Michael Rabbat** [1]

## Abstract

Agents that can reason and plan in the real world must be able to predict the consequences of their actions. World models possess this capability but require action annotations that can be complex to obtain at scale. Latent action models address this issue by learning an action space from videos alone. Our work studies the training of latent action world models on in-the-wild videos, expanding the scope of existing works that focus on simple robotics simulations, video games, or manipulation data. While diverse videos enable modeling richer actions, they introduce challenges of environmental noise and lack of a common embodiment across videos. To address these, we carefully study the design and evaluation of latent actions. We find that constrained continuous latent actions are better suited for complex in-the-wild videos, compared to vector quantization. For example, actions specific to in-the-wild videos such as humans entering the room, can be modeled and then transferred across videos. However, in the absence of a common embodiment, learned latent actions are localized in space, relative to the camera. Nonetheless, we are able to train a controller that maps known actions to latent ones, allowing us to use latent actions as a universal interface to solve planning tasks on par with action-conditioned baselines.

## 1. Introduction

To build intelligent systems that can reason and plan in the real world, we must endow them with the ability to predict the future and, in particular, the consequences of their actions (Friston, 2010; Clark, 2013; Bubic et al., 2010; LeCun, 2022; Sutton, 1991; Ha & Schmidhuber, 2018; Hafner et al., 2019; Nguyen & Widrow, 1990). When agents are

[1]FAIR at Meta [2]INRIA [3]New York University. Correspondence to: Quentin Garrido <garridoq99@gmail.com>.

*Proceedings of the 43rd International Conference on Machine Learning*, Seoul, South Korea. PMLR 306, 2026. Copyright 2026 by the author(s).

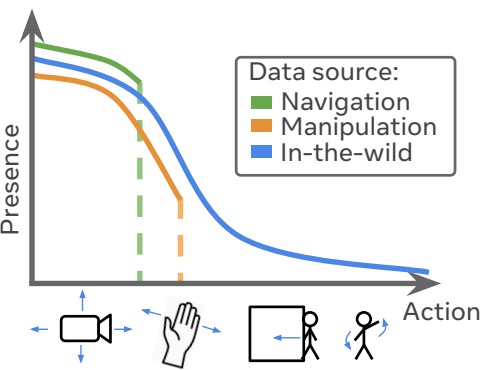

*Figure 1.* **Action diversity.** Classically used navigation or manipulation data contains the most general actions, such as camera or hand movements. In-the-wild videos extend this to a much broader distribution of actions, with people entering the scene or dancing.

present in the scene, predicting the future becomes inherently stochastic, parametrized by possible actions. Modeling these possible futures is thus necessary to learn good models of the world – ones that can, for example, be used for planning. A significant body of literature on world models assumes access to action labels (Ha & Schmidhuber, 2018; Hafner et al., 2023; Hu et al., 2023; Bar et al., 2024; Agarwal et al., 2025; Assran et al., 2025), which is a critical bottleneck: the vast majority of video data online is action-free (Zellers et al., 2022; Miech et al., 2019) and includes diverse embodiments.

This gap motivates the learning of latent action models (LAM) (Edwards et al., 2019; Rybkin et al., 2019; Menapace et al., 2022; Schmidt & Jiang, 2024; Ye et al., 2025; Yang et al., 2025; Chen et al., 2024; Cui et al., 2024) that can discover the action space from videos alone, without action annotations or a known embodiment. The standard approach is to learn two components jointly. First, an inverse dynamics model (IDM) that, given observations of the past and future, predicts a latent action that explains the difference between the two. Second, a forward model which predicts the future using the past and obtained latent action. After such a model is trained, the IDM can be frozen and used as part of a VLA pipeline (Bu et al., 2025; Ye et al., 2025) or to train a world model (Gao et al., 2025).

The type of videos used directly affects the learned action space, and is often an understudied component. Most LAM studies rely on narrow, task-aligned domains–video games (Bruce et al., 2024), tabletop manipulation (Nikulin

et al., 2025), or curated real manipulation (Bu et al., 2025; Gao et al., 2025)–which yield action spaces specialized to a single embodiment with limited transfer or generalization. While some works have used more "natural" videos from Ego4D (Grauman et al., 2022), it usually composes a small fraction of the training mix, e.g. 5% for Bu et al. (2025) and Gao et al. (2025), far from leveraging the richness of in-the-wild videos.

To learn a truly general and transferable latent action world model, we argue that we must go beyond these targeted data sources to natural in-the-wild videos such as HowTo100M (Miech et al., 2019) or YoutubeTemporal-1B (Zellers et al., 2022) which provide a much richer and general learning environment than usually studied, as illustrated in Figure 1. This introduces a new set of research challenges that we address in this work to demonstrate the viability of LAM on large scale in-the-wild natural videos[1]. First, the meaning of an "action" on in-the-wild video is not as clearly defined as it is in environments with agents having known action spaces. Considering the metaphorical principal component decomposition of actions, we could expect the *first component* to capture movements, something shared across video sources. From then we might have a split between ego- and exo-centric actions, which separates actions of the camera wearer and other agents in the environment. In in-the-wild videos, we have a stronger presence of external agents performing diverse actions, on top of what the camera wearer does. Going deeper in the action distribution, in-the-wild videos will contain unique actions such as cars entering the frame, people dancing, fingers forming chords on a fretboard, etc. This leads to an inherent richness of actions that we aim at modeling. In-the-wild videos provide a super-set of actions compared to video games or manipulation videos, which means that one should still be able to solve more classical navigation or manipulation tasks. While data sources used in previous works would mainly contain the *first principal components* of actions, trying to model more diverse actions has a risk of capturing more environmental noise (Nikulin et al., 2025) such as leaves oscillating on trees. Finally, agents in in-the-wild videos do not have a consistent embodiment that the model can latch onto, which poses challenges for transfer and downstream applicability of the learned latent actions. The focus of our work thus lies in the study of latent action world models trained on large scale in-the-wild video datasets, studying the inherent challenges, potential pitfalls of latent actions in such a setting, as well as demonstrating their viability for downstream planning.
Our contributions are as follows:

- We study how to regulate the information content of latent actions, focusing on in-the-wild natural videos. We find that sparse or noisy latent actions can effectively model complex actions, whereas discrete ones struggle to adapt.

- We show that the absence of a common embodiment across in-the-wild videos is not an issue when learning latent actions. Latent actions will encode more spatially-localized transformations.

- We demonstrate the generality of the learned action space by transferring complex actions between videos. We find that we can effectively transfer motion between objects, or actions such as someone entering the frame.

- We demonstrate how our learned latent action space can serve as a general action space. By training a small controller to map known actions to latent ones, our world model trained only on natural videos can be controlled to solve robotic manipulation and navigation tasks, achieving planning performance close to models trained on domain-specific, action-annotated data.

Overall, our work demonstrates the feasibility of learning a latent action conditioned world model purely using natural in-the-wild videos.

## 2. Related Work

**World Models.** While early world models focused on simple or game environments (Nguyen & Widrow, 1990; Ha & Schmidhuber, 2018; Hafner et al., 2019), recent work has scaled to complex real-world domains (Hu et al., 2023; Agarwal et al., 2025; Assran et al., 2025) to enable visual planning (Shah et al., 2021; Khazatsky et al., 2024). However, generalizing across diverse agents remains challenging; manual unification of action spaces is not scalable (Hansen et al., 2023). Latent Action Models (LAMs) address this by learning abstract action spaces directly from unlabeled video (Edwards et al., 2019; Rybkin et al., 2019).

**Latent Action Models.** LAMs (Edwards et al., 2019; Menapace et al., 2021; Schmidt & Jiang, 2024) typically employ an IDM to infer actions from state transitions. To prevent the IDM from trivially encoding the future (causal leakage), methods often rely on discretization (Bruce et al., 2024; Ye et al., 2025; Bu et al., 2025) or continuous regularization (Rybkin et al., 2019; Yang et al., 2025). However, many existing approaches operate in pixel space, making them sensitive to background distractors (Nikulin et al., 2025), or rely on two-stage training for the latent actions and world model (Yang et al., 2025). In contrast, we explore joint training of latent-space world models on large-scale in-the-wild videos, explicitly comparing continuous and discrete regularization strategies. Confer Appendix A for an extended discussion of related works.

---

[1]While our work does not focus on video generation, LAMs trained on in-the-wild videos could help reduce the text-video supervision burden (Sun et al., 2024).

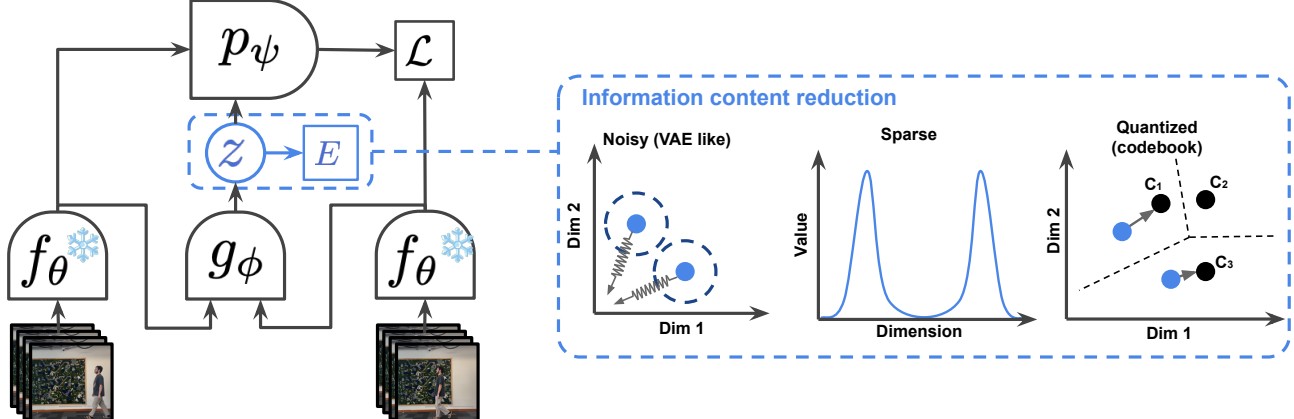

*Figure 2.* **Latent action world model.** A classical world model is endowed with actions represented as latent variables. A video is encoded through an encoder $f_\theta$, producing representations that are passed to the world model $p_\psi$ which predicts next frames. The latent actions $z$ are obtained thanks to an inverse dynamics model $g_\phi$ which can look into the future and is trained jointly with the world model. To limit the information content (and propensity to cheat) of latent actions, they are regularized using techniques such as noise addition, sparsification, or quantization.

## 3. Problem Setting

We want to model the stochastic evolution of the world state $s_t$ using a latent action formulation $s_{t+1} = p_\psi(s_{0:t}, z_t)$, where $z_t$ represents actions capturing the transition dynamics. To learn these actions from action-free videos, we use an IDM[2] $g_\phi$ that infers $z_t$ by looking at the past and future frame: $z_t = g_\phi(s_t, s_{t+1})$. The system is trained jointly to minimize the following prediction error

$$\mathcal{L}_t = \|s_{t+1} - p_\psi(s_{0:t}, z_t)\|_1 \text{, with } z_t = g_\phi(s_t, s_{t+1}).$$

In clean environments (Hoque et al., 2025; Yu et al., 2020), $z_t$ naturally aligns with agent actions. However, on in-the-wild videos (Zellers et al., 2022; Miech et al., 2019), the IDM risks encoding exogenous noise (e.g., leaves moving) (Nikulin et al., 2025). Therefore, limiting the information content of $z_t$ is paramount. We investigate three regularization strategies to enforce minimal explanations: sparsity, noise injection (VAE-like), and quantization.

**Sparsity.** The first one, and perhaps most complex to implement, is sparsity based constraints (Drozdov et al., 2024). Here, we would like for the latent actions to have as low of an $L_1$ norm as possible. A few additional regularizations are added due to trivial solutions that would reduce the $L_2$ norm of the vectors, concentrate the norm along a few dimensions, or focus too much around the mode of the latent distribution. Denoting the batch size by $B$ and dimension of the representations by $D$, the regularization is

$$\mathcal{L}(Z) = VCM(Z) + \frac{1}{N} \sum_i E(Z_i), \text{ with}$$

---

[2]We can see $z_t$ as the result of an optimization process minimizing the prediction error over it. Implementing it this way is impractical, but we can see the IDM as performing amortized inference (Amos et al., 2023).

$$E(z) = \lambda_{l2} \max\left(\sqrt{D} - \|z\|_2^2, 0\right) + \lambda_{l1}\|z\|_1, \text{ and}$$

$$VCM(Z) = \lambda_V \frac{1}{D} \sum_d \max\left(1 - \sqrt{\mathrm{Var}(Z_{\cdot,d})}, 0\right)$$
$$+ \lambda_C \frac{1}{D(D-1)} \sum_{i \neq j} \mathrm{Cov}(Z)_{i,j}^2 + \lambda_M \frac{1}{ND} \sum_{i,j} Z_{i,j}.$$

This Variance-Covariance-Mean (VCM) regularization, inspired by VICReg (Bardes et al., 2021), ensures an adequate spread of information and forces the sparsity constraints to be properly used by the model. In practice we set the coefficients to $\lambda_{l2} = 1$, $\lambda_V = 0.1$, $\lambda_C = 0.001$, $\lambda_M = 0.1$, and vary $\lambda_{l1}$ to regulate information content. We expand on the motivation behind this regularization and ablate each of its components in Appendix E.

**Noise addition.** Another approach to limit information content in the learned latent actions is to add noise to them, while making sure their norm does not increase and makes the noise negligible. This can be implemented in a similar way as a VAE (Kingma & Welling, 2014; Gao et al., 2025). The prior matching term here acts as our regularizer, where the target standard deviation adds noise while the target mean reduces the norm of the latent actions.

$$\mathcal{L}(z_t) = -\beta \, D_{KL}\left(q(z_t|s_t, s_{t+1})||\mathcal{N}(0, 1)\right)$$

**Discretization.** A final approach is to discretize the latent actions. For this, the most common approach is vector quantization (Van Den Oord et al., 2017) or a variant of it. This serves as a baseline comparison to illustrate a commonly used regularization in previous works (Ye et al., 2025; Bu et al., 2025). In practice, we use the same quantization scheme as UniVLA (Bu et al., 2025), using classical vector quantization (Van Den Oord et al., 2017) as well as codebook reset for unused codes.

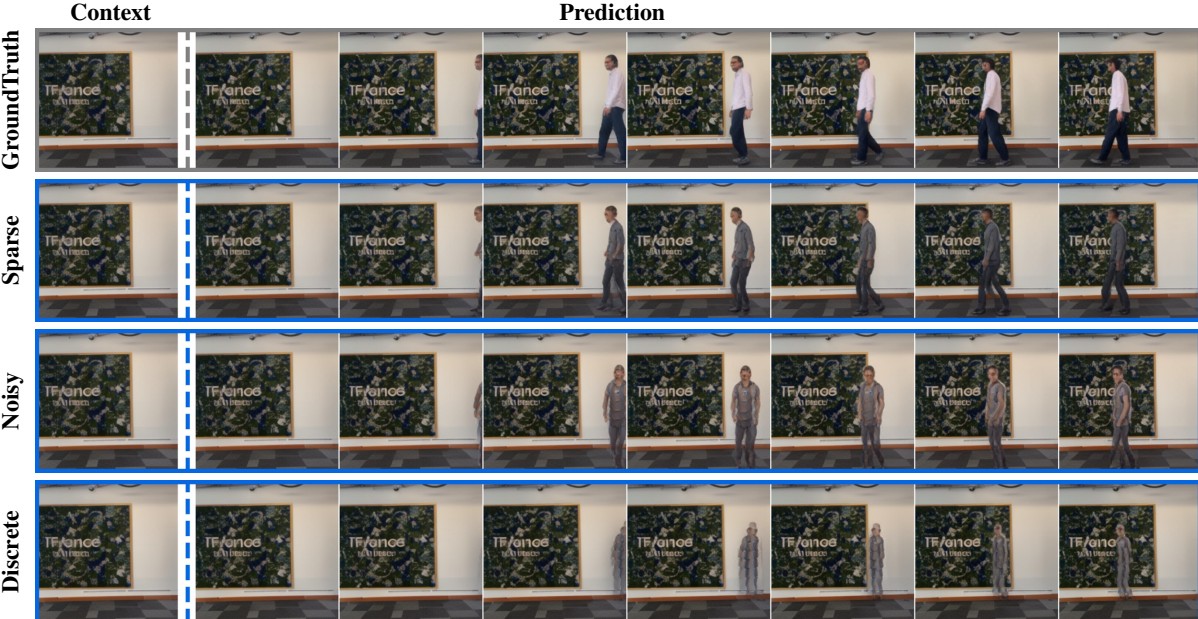

*Figure 3.* **Sample predictions using the IDM.** We illustrate the highest quality rollouts obtained with different regularization, using the inverse dynamics model. While sparse or noisy latent actions are able to capture a man entering the scene, discrete ones are not able to properly capture such action, even if some motion remains captured. Videos recorded by the authors.

All of this can be performed in the latent space of a trained encoder where $s_t$ and $s_{t+1}$ now are the representations obtained from video frames, which leads us to the complete architecture illustrated in Figure 2.

## 4. Experimental details

We now turn ourselves to a more practical implementation. A video $V$ of length $T$ is encoded through a frame-causal encoder $f_\theta$ –V-JEPA 2-L (Assran et al., 2025) in our experiments– producing representations $s_{0:T-1}$. This encoder is kept frozen during training. We then train the world model $p_\psi(s_{0:t}, z_t)$ and inverse dynamics model $g_\phi$ jointly to predict $s_{t+1}$ using the aforementioned prediction loss and latent action regularization.

For efficiency, we train the model using teacher forcing (Williams & Zipser, 1989; Vaswani et al., 2017). By default, $p_\psi$ is implemented as a ViT-L (Dosovitskiy et al., 2021) using RoPE (Su et al., 2021; Assran et al., 2025) for positional embeddings. To condition $p_\psi$ on $z$ we use AdaLN-zero (Peebles & Xie, 2023) that we adapt to condition the sequence frame-wise. Our latent actions $z_t$ are 128 dimensional continuous vectors. For our IDM $g_\phi$, we use two self-attention ViT blocks applied to pairs of subsequent $(t, t + 1)$ frames, followed by a cross-attention block with a learnable query that is processed into our latent action $z_t$. Unless specified otherwise, all models are trained on YoutubeTemporal-1B (Zellers et al., 2022) using the Muon optimizer (Jordan et al., 2024).

For visualization, we train a frame-causal video decoder us-

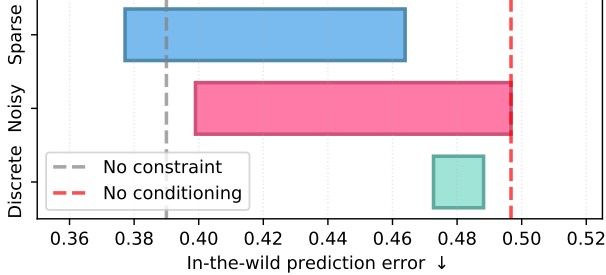

*Figure 4.* **IDM performance.** We report the one step prediction error on in-the-wild videos. Adjusting the capacity of sparsity and noise based latent actions allows for varying performance, while quantized ones struggle to adapt to the complexity.

ing a ViT-L trained with a combination of $L_1$ and perceptual loss (Johnson et al., 2016; Zhang et al., 2018). While generation is not core to our work, this is a useful tool to compute perceptual metrics and inspect the model's predictions. See Appendix B for detailed protocols and hyperparameters. We further explore the use of a different encoder, namely DINOv3 (Siméoni et al., 2025) in Appendix D, where we find similar findings as when using V-JEPA 2.

## 5. Performance of information regularizations

As aforementioned, we want to capture rich and complex actions that span a wide range of embodiments, as observed in the in-the-wild videos we consider. The first question we thus want to answer is how different information regularization techniques adapt to this complexity?

While we measure performance in various manners through

**a)** 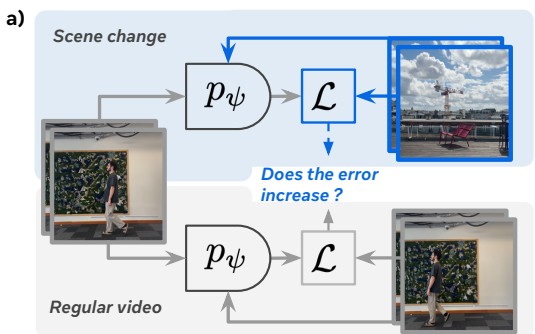

**b)** 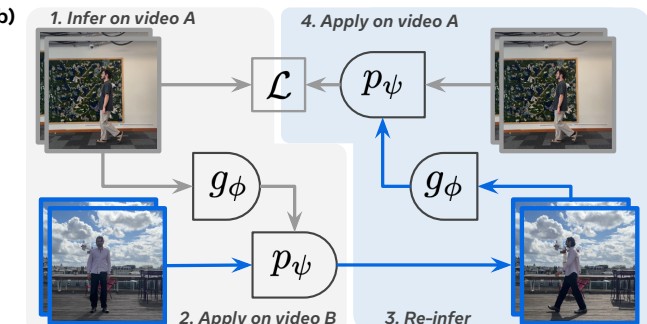

*Figure 5.* **Raw latent evaluation.** By artificially stitching videos, we can create abrupt scene changes. Measuring how the prediction error increases when such changes happen compared to the original video tells us how well the model can capture the whole next frame **(a)**. To measure the transferability of latent actions, we measure if their inference is cycle-consistent. We infer latent actions on video A, then apply them to another video. From this prediction, we re-infer the latent actions and apply them on video A. If the latent action transfers well, we should obtain a small error with video A **(b)**. The use of both metrics ensures that shortcuts are not the source of the transfer.

the remainder of the manuscript, focusing on different aspects and properties, we first examine the prediction quality in an ideal setting. Here we will measure the prediction error of models when rolling out a trajectory, using the inverse dynamics model (and thus the future frame) to infer the actions. This gives an upper bound of performance across all other experiments.

We will say that a regularization is "better" if it leads to a variety of achievable performance and does not saturate easily. Being able to explore a multitude of behaviors also enables us to measure the impact of latent capacity on downstream performance. As we show in a later section, achieving the lowest prediction error using the inverse dynamics model is not always desirable, as downstream tasks require a balance between complexity and identifiability of latent actions. As we can see in Figure 4, sparse and noisy latent actions are able to achieve a range of performance between unconstrained latent actions (using the whole continuous space) and a deterministic world model. Even at maximal sparsity, we still have $d = 128$ latent actions with sparsity constraints, where when the weight $\beta$ of $D_{KL}$ becomes high, noisy latent actions become noise, equivalent to no conditioning. However, discrete latent actions struggle to scale their capacity and remain close to the deterministic baseline. We provide further analyses on the failure of the discrete latent actions in Appendix F.

In the rest of this work, we talk about this "in-the-wild prediction error" as capacity of the latent actions. Since everything else in the training is identical, the drop in prediction error is attributed to the capacity of the latent actions. Lower prediction error indicates higher capacity latent actions, while a higher one indicates lower capacity latent actions. Qualitatively, we can see in Figure 3 that continuous latent actions can capture complex object dynamics such as someone entering the room, whereas discrete latent actions struggle to model it accurately. Confer Appendix K for additional visualizations.

*Table 1.* **Prediction error increase under scene changes.** On Kinetics (Kay et al., 2017), all models exhibit a significantly higher error when a scene change occurs. This shows that the latent actions cannot simply copy the next frame. We report LPIPS values for ease of interpretation.

| Latents | Capacity | w/o change | w/ change |
|---|---|---|---|
| Sparse | Low | 0.28 | 0.66 ($\times 2.3$) |
| | High | 0.20 | 0.50 ($\times 2.4$) |
| Noisy | Low | 0.33 | 0.69 ($\times 2.1$) |
| | High | 0.21 | 0.54 ($\times 2.5$) |
| Discrete | Low | 0.34 | 0.69 ($\times 2.0$) |
| | High | 0.29 | 0.68 ($\times 2.3$) |

## 6. What kind of actions do we learn?

While we used an ideal setting where latent actions are inferred by the IDM, the model could simply cheat and encode the next frame in the latent action. Or we could learn latent actions that cannot be applied on another video, contrary to our goal of them being *minimal explanations*. We now study these two problems with simple and intuitive metrics. See Figure 5 for illustration of the protocols.

**Do latent actions simply encode the future?** We quantify information leakage by measuring prediction error across artificial scene cuts. If the latent action $z_t$ simply encodes the target frame $s_{t+1}$ (cheating), reconstruction error will remain low despite the context discontinuity. As shown in Table 1, all models exhibit a sharp increase in error on scene cuts, confirming that the learned latents do not trivially copy the future frame. We hypothesize that the complexity of the used dataset makes it harder for the model to learn this solution. Visual inspection in Supplementary Figure S1 reveals that while some information about the next frame is captured in the latent actions, it is minor. Nonetheless, as we study in transferability evaluations, this is not an issue in practice, and likely to be a consequence of having to encode objects appearing in and out of frames.

**Context**                    **Prediction**

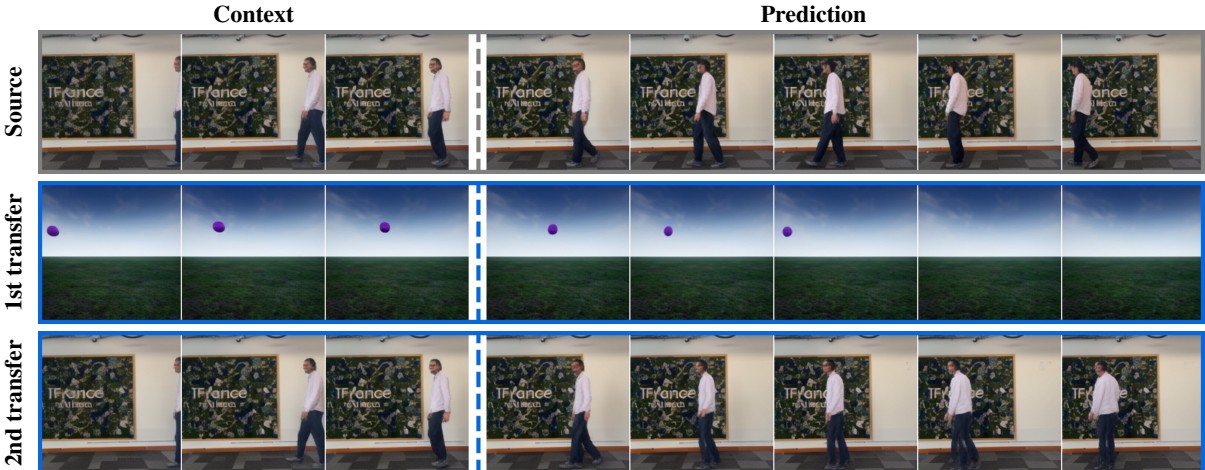

*Figure 6.* **Transfer and cycle consistency.** We infer latent actions from a source video, here of a man moving to the left. We then apply these actions to a flying ball, which stops its motion and also starts moving left, demonstrating transferability of latent actions. We then re-infer the latent actions and apply them to the original video. We can see the man moving to the left again, indicating that the motion was re-inferred correctly. Human videos recorded by the authors, flying ball video from (Riochet et al., 2022).

*Table 2.* **Action cycle consistency.** Actions are inferred on Video 1, then applied on Video 2. Actions are again inferred and applied again on Video 1. The small increase in prediction error indicates that actions can reliably be transferred and re-inferred. We report LPIPS values over 2s prediction for ease of interpretability.

| Latents | Capacity | Kinetics | | RECON | |
|---|---|---|---|---|---|
| | | Original | Transfer | Original | Transfer |
| Sparse | Low | 0.26 | 0.31 (×1.20) | 0.24 | 0.29 (×1.21) |
| | High | 0.19 | 0.24 (×1.30) | 0.20 | 0.23 (×1.14) |
| Noisy | Low | 0.30 | 0.34 (×1.13) | 0.29 | 0.33 (×1.15) |
| | High | 0.20 | 0.26 (×1.34) | 0.20 | 0.24 (×1.22) |
| Discrete | Low | 0.32 | 0.33 (×1.03) | 0.32 | 0.33 (×1.03) |
| | High | 0.27 | 0.29 (×1.07) | 0.26 | 0.27 (×1.05) |

**Do latent actions transfer well?** The next experiment checks whether we can apply latent actions inferred on a video to another one. Quantitatively, we evaluate the models on cycle consistency of latent actions. From random videos A and B, we infer latent actions on video A then apply them on video B. If the latent actions transfer well, we should be able to infer them again. We thus infer them again on video B and apply them on video A. By measuring the increase in prediction error on video A with the original and cyclically inferred latent actions, we can see how well latent actions transfer. While this transfer is not well-defined on random natural videos, leading to absolute gaps that are hard to interpret, this can still allow us to rank models and get an intuition about this transfer. We can see in Table 2 that on both Kinetics (Kay et al., 2017) (human activity videos) and RECON (Shah et al., 2021) (navigation) we only obtain a minor increase in prediction error over this latent inference cycle. While latent actions with higher capacity lead to a worse transfer, their performance remains higher after transfer than their more constrained counterparts. As shown by the previous lack of leakage of the future frame, this transfer does not stem from copying the next frame, which

would be a way to obtain perfect performance. Qualitative experiments in Figure 6 confirm that our latent actions are robust enough to enable motion transfer between disparate embodiments, while retaining cycle consistency. Confer Appendix L for additional visualizations.

**Which embodiment do the latent actions learn?** Visualizations in Figure 7 reveal that the learned latent actions encode spatially localized, camera-relative transformations rather than semantic object-centric ones, explaining the previously observed transferability. This camera-relative embodiment can be a strength as it allows us to transfer motion between entirely different objects, which would not be possible if motion only targeted semantically similar objects.

## 7. Latent action world models and planning

One application of a latent action space is to use it as a generic interface for various embodiments. By learning a mapping from "real" actions to latent ones, we can control the world model in an interpretable way. This also allows us to solve planning tasks, as we study in this section.

**Controller training.** The first part is to train a module to go from real actions – and optional representations – to latent actions. In the case of using actions alone, we use a simple MLP, and when using actions and past representations, we use a cross-attention based adapter. Confer Appendix B for detailed architecture and protocols. We then simply train this controller module to predict the latent action with an $L_2$ loss. We illustrate this process in Figure 8. Due to the learned latent actions being camera relative, using actions alone can be insufficient as the target latent actions will vary not only based on the action but also camera position. In practice, we find that the controller converges to a la-

**Animating the right person**      **Animating the left person**

*Figure 7.* **Action locality.** We apply a localized locomotion action to a video with two individuals inside of it. We find that only the person closest to the walking man in the first video starts moving, indicating that the action has localized properties. We are making the individual at a given position move to the left. Videos recorded by the authors.

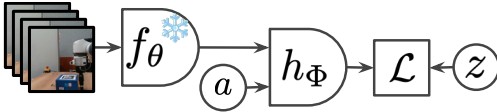

*Figure 8.* **Controller training.** We train a lightweight module to map known actions to latent actions. Representations of the past are used to help the prediction of the right latent actions.

tent action that leads to no movement when not using past representations. Confer Appendix M for visualizations.

**Rollout quality.** We train controllers on DROID (Khazatsky et al., 2024), a robotic manipulation dataset, as well as RECON (Shah et al., 2021), a navigation dataset. DROID allows us to evaluate the model on data where the camera is fixed but an agent is moving inside of the scene, while RECON has still scenes but where the camera wearer is the one moving. As we can see qualitatively in Figure 9 and quantitatively in the left subplots of Figure 10, models are able to achieve quality predictions when using the controller. The predictions obtained when using the controller are very similar to the ones obtained with the IDM, with slightly more conservative actions.

We however find a lack of correlation between the prediction error on in-the-wild videos, i.e., the capacity of the latent actions, and the quality of the rollouts when using the controller. For both sparse and noisy latent actions, we find that using the most or least constrained setting is suboptimal, and that a more balanced regularization leads to the best predictions. This can intuitively be explained by over-constrained latent actions not containing enough information, and under-constrained ones containing too much information about the future. This is consistent with the trends observed previously, where more constrained latent actions transfer better, but freer ones can capture more fine-grained motion. Due to the simplicity of the action space here, we see that even discrete latent actions work well, supporting this choice in prior work (Bu et al., 2025; Schmidt & Jiang, 2024). Confer Appendix H for detailed results.

**Planning.** We can now use our trained controllers and measure performance on goal-based planning tasks using

existing protocols. Given an initial observation $s_t$ and goal observation $s_g$, we seek an action sequence that minimizes the distance between the predicted and goal states.

For our DROID controller, we adopt the protocol of Terver et al. (2026) and use a set of videos recorded in the real world on a Franka Emika Panda. We consider trajectories where the goal is to move the arm to a specific goal position. We plan at a horizon of $H=3$ steps using the Cross-Entropy Method (CEM) (Rubinstein, 1997) and compare ourselves to the performance of V-JEPA 2-AC which is trained in a similar way as our model but using known actions, as well as the best model based on V-JEPA 2 from Terver et al. (2026) to upper bound the performance. To measure performance, we use the distance to the goal ($\Delta xyz$) which can be easily computed thanks to the compositionality of translations. Confer Appendix B for the detailed protocol. While performance remains lower than specifically designed models, our models are able to achieve similar performance to V-JEPA 2-AC, demonstrating that our learned latent actions can effectively be used as an interface for planning tasks. Here, the higher capacity latent actions, even though they may produce worse rollouts, can lead to the best planning performance. Notably, noisy latent actions obtain the best planning performance when the rollouts are suboptimal. We explore the impact of adding domain specific data in our pipeline in Appendix I.

On a navigation task, using our controller trained on RE-CON, we follow the protocol of NWM (Bar et al., 2024) and evaluate performance using CEM for planning. We rely on the Relative Pose Error (RPE) (Sturm et al., 2012) between planned and ground truth trajectories as our main metric. We find similar conclusions here with models able to achieve performance that, while not on par with NWM, are able to beat policy based baselines such as NoMaD (Sridhar et al., 2024). Egocentric navigation has the added difficulty of additional information entering the frame at every prediction step, making it harder to produce clean rollouts and lowering performance. For more detailed planning results, confer Appendix H. Nonetheless, we find that the quality of the rollout is not perfectly correlated with planning per-

**DROID**                                    **RECON**

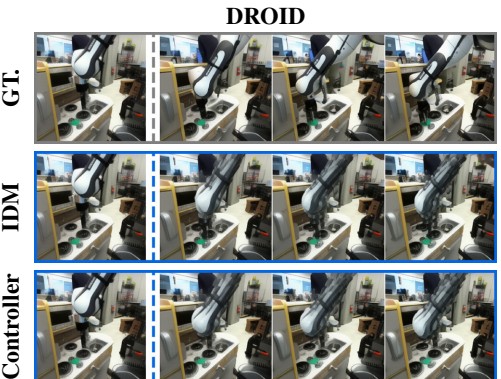
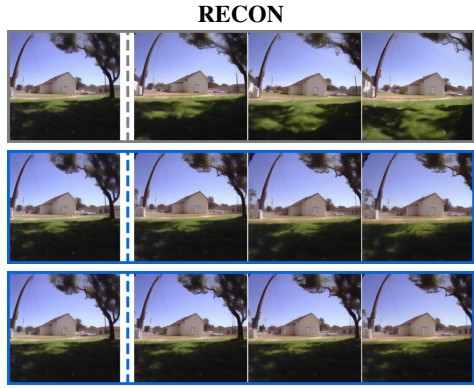

*Figure 9.* **Rollouts using the controller and IDM.** On both DROID and RECON, the controller is able to approximate the latent action produced by the inverse dynamics model. Movements are applied correctly over the rollout, however physical appearance degrades over time. To produce the rollouts, frames are duplicated to map one action to one latent, something not seen during training.

**DROID**                                    **RECON**

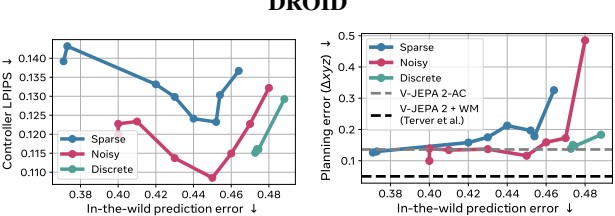
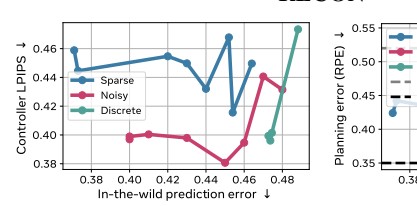
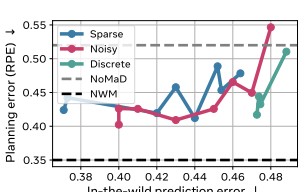

*Figure 10.* **Controller and planning performance.** On both DROID and RECON, we are able to successfully train a model to map real to latent actions (left plots). Using these actions with classical planning protocols, we are able to achieve similar performance to world model or policy baselines, that are trained with actions from the start (right plots). Overall, the best performing models are the ones where the latent actions form a middle ground in terms of capacity.

formance. This is a common challenge in the world model literature(Zhang et al., 2025).

Overall, we find that our models trained only on in-the-wild videos learn latent action spaces that can effectively be reused to solve simple planning problems, while never having seen this data during training.

## 8. Scaling models and data

In this section we investigate how the performance of the models scales as we increase data, model size, and training time. We focus on sparse (with $\lambda_{l1} = 0.01$) and noisy latent actions (with $\beta = 5 \times 10^{-5}$). Looking at both allows us to study scaling trends in diverse settings. We can see in Figure 11 that overall, as model size, training time, or training data increase, we obtain better predictions when using the IDM on natural videos. However, looking at the planning performance on DROID shows us a more nuanced story, where training time significantly improves the performance but model size mainly has an effect for the noisy latent actions, and training data does not show a significant trend. This nuanced story about model size is consistent with previous work (Ye et al., 2025) which also find minor increase in performance when performing scaling analyses. These results suggest that although scaling can improve the quality

of a latent action world model by improving the quality of the latent actions and/or forward model, this may not always be visible in downstream tasks that mainly evaluate simple actions, as are often used in the literature.

## 9. Limitations and future work

**Variable latent action constraints.** Our current regularization placed on latent actions is constant, which does not account for the varying action complexity of videos. Even though it may increase the complexity of the latent action space, it would be interesting to dynamically adjust the constraint based on the complexity of the video.

**Sampling and planning.** While we analyzed the transfer of latent actions as well as their use as a control interface, we did not exploit the latent actions directly for sampling or planning (Rybkin et al., 2019), which would more directly assess their quality. We provide some initial analysis on these aspects in Appendix G, but most of the work is ahead for high dimensional structured latent actions.

**Shaping representations with single stage training.** Relying on frozen representations limits the quality of predictions as this space was not optimized for it. As we use similar data to V-JEPA 2 in our work, the use of latent actions in a V-JEPA 2-style pretraining could unlock a symbiotic

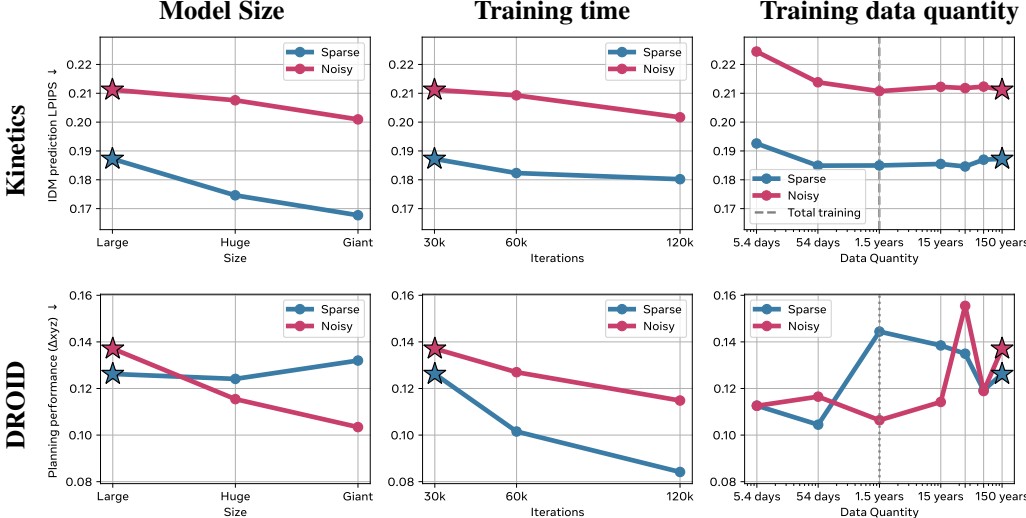

*Figure 11.* **Scaling trends.** We investigate for two sets of latent regularizations the performance behavior when scaling the model size (left), total training time (middle), and training data quantity (right). We find that for all axes of scaling, we are able to obtain an improved IDM on natural videos (top row). We see that when measuring performance on planning tasks we obtain similar trends, with the clearest improvements obtained by training longer. (bottom row). For data scaling, we note that our usual recipe sees on average every video twice, but we only see a total of 1% of the total number of frames. This latter number is when we start to see degraded performance due to a too small training set. Stars indicate our default setup in the rest of the paper.

single-stage encoder/world-model training.

## 10. Conclusion

This work demonstrates the feasibility of learning effective Latent Action World Models directly from large-scale, in-the-wild natural video datasets. We successfully address the significant challenges posed by this data, including high action complexity, environmental noise, and the lack of a common embodiment. Our study of information regularizations highlights the benefit of continuous latent actions, which are able to adapt more effectively to the complexity of actions present in in-the-wild videos. Vector quantization, although very common in practice, struggles to adapt to this scale. By studying the leakage of future frames in the latent actions, we found that this problem is not present in a practical setting, which we hypothesize is due to a combination of conditioning choice and data complexity. We further found that while higher capacity latent actions hurt transferability, latent actions were still able to be inferred and reapplied consistently. This led to the finding that, on natural videos, learned latent actions are spatially-localized relative to the camera due to the lack of a common embodiment across videos. Qualitatively, the learned latent actions can capture complex actions, such as a person entering a scene, and can even transfer motion between different objects, such as from a human to a ball. Most critically, we demonstrated the practical utility of this approach. By training a simple controller to map state and known actions to the learned latent actions, our world model—trained exclusively on in-the-wild, natural videos—can be controlled to solve robotic manipulation tasks. It achieves planning performance comparable to baselines trained on in-domain, action-annotated data. Overall, our analyses and experiments demonstrate the viability and potential of training latent action models on in-the-wild natural videos, offering a step towards more general world models.

## Acknowledgments

We would like to thank Adrien Bardes for accepting to act in videos used for qualitative results, as well as for fruitful discussions. We also thank Amir Bar for discussions and advice on planning experiments.

## Impact Statement

This work makes progress towards autonomous agents that can learn from large-scale in-the-wild video data, potentially accelerating progress in robotics and physical reasoning. However, by relying on in-the-wild data sources such as YoutubeTemporal-1B, our learned world models may inherit biases or stereotypes present in the training distribution. Furthermore, even though our primary application is in solving planning tasks, the studied methods here could be applied to controllable video generation. This can raise issues pertaining to synthetic video generation and video manipulation. These risks are however mitigated due to the optimization and development of our models for control tasks rather than visual fidelity.

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

# A. Extended related works

**World Models.** World Models (Nguyen & Widrow, 1990; Sutton, 1991; Ha & Schmidhuber, 2018) have become a very active area of research. While a significant body of work had been applied to game data (Alonso et al., 2024; Hafner et al., 2019; 2023), applications to more complex environments, such as simulated robotics environment (Seo et al., 2023; Zhou et al., 2024) or the real world (Hu et al., 2023; Agarwal et al., 2025; Assran et al., 2025) have flourished recently. With a plethora of possible embodiments and action space, works such as NWM (Bar et al., 2024) focus on locomotion, PEVA (Bai et al., 2025) on whole body control, or UniSim (Yang et al., 2023) which can handle a variety of embodiments through textual control, have appeared. The promise of such models is not solely to generate visually appealing videos (Brooks et al., 2024; Teng et al., 2025; Agarwal et al., 2025) but mainly lies in their use to solve visual planning tasks. Being able to predict the consequences of actions can enable us to solve problems for navigation (Shah et al., 2021), robotic manipulation in simulation (Nasiriany et al., 2024; Liu et al., 2023; Yu et al., 2020) or in the real world (Khazatsky et al., 2024), or even whole body control (Ma et al., 2024). Such models can even be used to solve more classical vision tasks such as segmentation and depth forecasting (Baldassarre et al., 2025; Karypidis et al., 2024; Luc et al., 2017). A common issue to obtain models that generalize across embodiments is how to define a common action space? A solution can for example be to use the maximal dimensionality across considered embodiments, with an embodiment token (Hansen et al., 2023), but this is not easily scalable. This is where latent action models (Edwards et al., 2019; Rybkin et al., 2019; Schmidt & Jiang, 2024; Bruce et al., 2024) come into play, as one of their promises is to learn an abstract, general latent action space.

**Latent Action Models.** Latent action models aim at learning actions from unlabeled videos. Latent actions can be inferred using a latent policy (Edwards et al., 2019), or by using an explicit IDM that predicts the latent action from the past and future frames (Rybkin et al., 2019; Menapace et al., 2021; 2022; Schmidt & Jiang, 2024). This is then combined with a forward model that predicts the future frame from the past and the latent action. The use of an IDM introduces a causal leakage in information and a key challenge is to ensure that the latent actions do not capture too much information, e.g. the entire next frame. A commonly used approach is to discretize the latent actions. This is the approach of choice in methods such as ILPO Edwards et al. (2019), LAPO (Schmidt & Jiang, 2024), Genie (Bruce et al., 2024), LAPA (Ye et al., 2025), or UniVLA (Bu et al., 2025). This can for example be motivated by prior knowledge of the desired action space (Bruce et al., 2024). Other methods such as CLASP (Rybkin et al., 2019), CoMo (Yang et al., 2025), or AdaWorld (Gao et al., 2025) instead opt for a continuous space, which is inherently more flexible. In this case, a regularization term can be added to reduce the information content of the latent actions. In between, PlaySlot (Villar-Corrales & Behnke, 2025) proposes to use both quantized latents as well as their residuals. Other works instead rely on carefully designed forward model architectures Menapace et al. (2022); Sun et al. (2024) to structure the latent action space. Furthermore, while numerous methods use off-the-shelf vision encoders to encode frames, latent actions are still often learned by predicting the future frame in pixel space (Chen et al., 2025; Yang et al., 2025; Ye et al., 2025). This makes latent actions more susceptible to distractors (Nikulin et al., 2025), where the latent actions learn to encode background noise rather than the actions we desire. While a solution is to use supervision (Nikulin et al., 2025; Liang et al., 2025), working in an abstract latent spaces and carefully designing latent actions can help avoid some of these issues, as we study throughout our work. In general, while learning latent actions has clear applicability to world models, methods tend to be developed with VLAs in mind (Bu et al., 2025; Ye et al., 2025). Even if the approaches are architecturally similar to world models, where the forward model/action decoder can be seen as a world model, it is often discarded. Even when a world model is trained, a two-stage approach is commonly used, where the world model is trained after the inverse dynamics model (Yang et al., 2025). Concurrently to our work Wang et al. (2025) proposes to treat the forward model as a world model, by using a pretrained video generation model.

# B. Training and evaluation protocols

**Decoder training.** Our decoder is trained using a ViT-L (Dosovitskiy et al., 2021) architecture, using RoPE (Su et al., 2021; Assran et al., 2025) positional embeddings. It reuses the architecture of the V-JEPA 2 encoder (Assran et al., 2025), with an added linear layer to map from patch to pixels. The decoder processes the full video sequence with a frame-causal attention mask to only attend to past frames.

It is trained using a combination of $L_1$ and perceptual loss (Johnson et al., 2016; Zhang et al., 2018). The decoder's weights are optimized using the Muon optimizer, with a learning rate of $0.02$, AdamW learning rate of $3 \times 10^{-4}$ and weight decay of $0.01$. We train the model with a batch size of 512, for 90 000 iterations, using a linear learning rate warmup for 12 000 iterations, followed by a cosine annealing.

**Latent action training.** By default, our world model $p_\psi$ uses a ViT-L (Dosovitskiy et al., 2021) architecture equipped

with RoPE (Su et al., 2021; Assran et al., 2025) positional embeddings. We condition $p_\psi$ on latent actions $z$ through an adapted AdaLN-zero (Peebles & Xie, 2023) mechanism that performs frame-wise conditioning, instead of the original sequence-wise conditioning. Each latent action $z_t$ is represented as a 128-dimensional continuous vector. The IDM $g_\phi$ uses two self-attention ViT blocks applied to pairs of subsequent $(t, t+1)$ frames, followed by a cross-attention block with a learnable query that is processed into the latent action $z_t$. We train the world model for next frame prediction using teacher forcing (Williams & Zipser, 1989; Vaswani et al., 2017) for computational efficiency.

We train on YoutubeTemporal-1B (Zellers et al., 2022) with batches of size 1024 for $30\,000$ iterations. For optimization, we rely on the Muon optimizer (Jordan et al., 2024) with a learning rate $0.02$ alongside AdamW (Loshchilov & Hutter, 2019) at a learning rate of $6.25 \times 10^{-4}$. The learning rate schedule begins with a linear warmup for the first 10% of training iterations, followed by cosine annealing. Weight decay is set to $0.04$. Training takes approximately 12 hours on 64 H100 GPUs.

The training loss can be defined as

$$\mathcal{L}_t = \|s_{t+1} - p_\psi(s_{0:t}, z_t)\|_1 + \mathcal{L}_z(z_t) \text{, with } z_t = g_\phi(s_t, s_{t+1}),$$

with $p_\psi$ the world model, $s_{0:t}$ is the sequence of past representations (encoded frames), $z_t$ the latent action inferred by the inverse dynamics model $g_\phi$ from consecutive representations $s_t$ and $s_{t+1}$, and $\mathcal{L}_z$ the regularization applied to the latent action.

To determine the coefficient used for the latent action regularization terms, we perform a sweep by increasing and decreasing the coefficients regulating information content until the latent actions have the same effect as noise, until an increase in capacity does not yield a reduction in prediction error, or for vector quantization when the codebook starts to not be fully utilized. This leads to the following coefficients:

- **Sparsity**: $\lambda_{l2} = 1$, $\lambda_V = 0.1$, $\lambda_C = 0.001$, $\lambda_M = 0.1$, $\lambda_{l1} \in \{0.4, 0.1, 0.08, 0.06, 0.05, 0.04, 0.02, 0.01\}$

- **Noisiness**: $\beta \in \{5 \times 10^{-3}, 1 \times 10^{-3}, 5 \times 10^{-4}, 1 \times 10^{-4}, 5 \times 10^{-5}, 1 \times 10^{-5}, 5 \times 10^{-6}, 1 \times 10^{-6}\}$

- **Discretization**: Commitment loss coefficient $\beta = 0.25$, $|C| \in \{16, 1024, 4096, 32768\}$, codebook reset for unused codes every 300k videos seen, equivalent to 2.5 million latent actions produced

**Controller training.** Our controllers consist of 2 self-attention blocks used to process the representation of the previous frame (we only look at the ultimate previous frame $s_{t-1}$, not the whole past $s_{0:t-1}$) followed by a cross-attention block between embedded real actions, and processed representations. This architecture is almost identical to the IDM one. Actions are embedded with a 3-layer MLP to a target embedding dimension chosen as the same as the encoder (1024 by default). The output singular token per timestep is then projected to the latent action dimension of 128 with a linear layer.

Since our latent action world models are trained with one latent action for two frames due to the video tokenization, we duplicate frames in the dataset to obtain a clear one-to-one mapping between real and latent actions.

The controller is then trained for 3000 iterations using the AdamW optimizer (Loshchilov & Hutter, 2019), with a learning rate of $1 \times 10^{-3}$, a weight decay of $0.04$, $\beta_1 = 0.9$ and $\beta_2 = 0.999$. The learning rate follows a linear warmup for 300 iterations and then a cosine decay for the rest of the training. We use a batch size of 256 with 8-frame videos at 4fps (which gives us 16 frames after duplication).

**Planning protocol for DROID.** Our model is used for planning using the protocol of Terver et al. (2026), which is as follows. Let $s_t = f_\theta(V_t)$ denote the latent visual state obtained by encoding the frame $V_t$ through the encoder $f_\theta$. Given an initial observation $s_t$ and a goal observation $s_g$, we seek an action sequence $a_{t:t+H-1} := a_t, \ldots, a_{t+H-1}$ that leads from $s_t$ towards $s_g$ over a planning horizon $H$. In practice, we use $H = 3$

We define the planning cost of an action sequence as

$$C(s_t, a_{t:t+H-1}, s_g) = \|s_g - \hat{s}_{t+H}\|_2, \tag{1}$$

where $s_g = f_\theta(V_g)$ is the encoded goal state, and the predicted latent visual states $\hat{s}$ are obtained by recursively rolling out the predictor:

$$\hat{s}_t = f_\theta(V_t), \quad \hat{s}_{i+1} = p_\psi(\hat{s}_i, c(a_i, \hat{s}_i)), \quad i \in [t, t+H-1], \tag{2}$$

with $c$ denoting the controller that maps actions and latent visual states to latent actions.

We use the Cross-Entropy Method (CEM) (Rubinstein, 1997) to solve this optimization problem. CEM maintains a Gaussian distribution over action sequences, initialized with zero mean and unit variance. At each iteration, we sample $N = 300$ candidate action sequences from the current distribution, evaluate their costs using the world model, and refit the distribution to the top $K = 10$ elite samples. We perform $I = 15$ iterations of this procedure and select the first action of the best sequence for execution.

To evaluate planning performance, we run 64 independent episodes. For each episode, we randomly select one video from 16 validation videos and randomly sample a clip of $H + 1 = 4$ frames at 4 fps (matching training conditions). We then define our error as the distance to the goal, defined as the $L_1$ distance between the cumulative planned actions and the cumulative groundtruth actions from the dataset:

$$\Delta xyz = \left\| \sum_{i=t}^{t+H-1} a_i^{\text{plan}} - \sum_{i=t}^{t+H-1} a_i^{\text{gt}} \right\|_1, \tag{3}$$

where $a_i^{\text{plan}}$ denotes the planned action at timestep $i$ and $a_i^{\text{gt}}$ the corresponding groundtruth action leading from $s_t$ to $s_g$. This metric measures the difference in total displacement between the planned and groundtruth trajectories, which is well-suited for actions that are additive in time, since multiple (infinitely many) paths can lead to the target. We report the error averaged across all 64 episodes.

**Planning protocol for RECON.** We use a similar protocol as for DROID, following the exact one used by NWM (Bar et al., 2024) which we recall for clarity. For additional details, confer Bar et al. (2024). Here for the Cross Entropy Method, we use $N = 120$ candidate actions and only a singular iteration, which was found to be sufficient in NWM.

For efficiency, trajectories are assumed to be straight lines, which allows us to plan only a single action that can be divided into the right number of time-steps. The planning horizon is here $H = 8$ which at 4fps represents 2 seconds in the future.

Once the trajectory is planned, we can compute the Absolute Trajectory Error (ATE) and Relative Pose Error (RPE) (Bar et al., 2024; Sturm et al., 2012) to measure the quality of the trajectory compared to the groundtruth ones. In practice we focus on RPE in the main body of our work, but ATE results are reported in Appendix H.

## C. Qualitative analysis of future leakage under scene changes

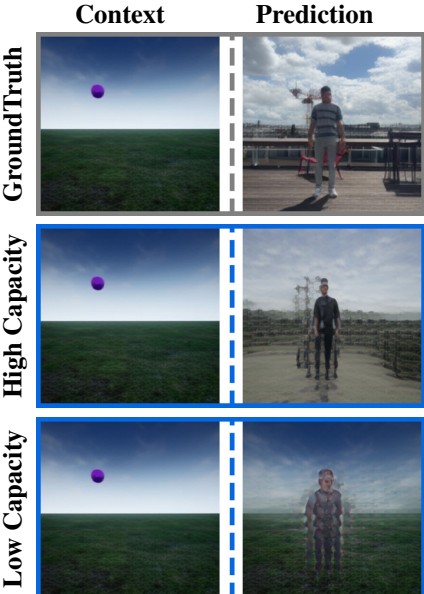

*Figure S1.* **Future leakage.** In the presence of a scene cut, the only solution is for the latent action to encode the next frame. As the capacity of the latent actions increase, more of the scene can be reconstructed, albeit with an extremely poor quality.

In this section, we take a qualitative look at how well latent actions can reconstruct the next frame in the case of scene change. We consider an extreme case where a flying ball switched to a human on a rooftop. In this case, the only solution is to encode as much as possible of the next frame. We compare latent actions with low and high capacity, corresponding respectively to $\lambda_{l1} = 0.4$ and $\lambda_{l1} = 0.01$. We can see that even with a high capacity we get a very poor reconstruction of the future frame, where the person is the clearest predicted aspect. With low capacity, the model is even unable to change the background and we only see the ball disappear and a human appear. This suggests that the latent actions focus better on humans, which are a main source of motion in videos. It is also worth noting that doing well in the case of a scene change does not mean that the model always cheats, but that it has enough capacity to encode parts of the future frame.

# D. Impact of encoder choice on behavior and performance

While we focused on V-JEPA 2 as our base encoder in the main body of this work, we here consider another model, DINOv3 (Siméoni et al., 2025) to study whether or not our findings change with a different encoder. We use encoders of similar size (ViT-L), using the same training and evaluation protocol. For this study, we focus on noisy latents, and mainly study downstream performance on DROID.

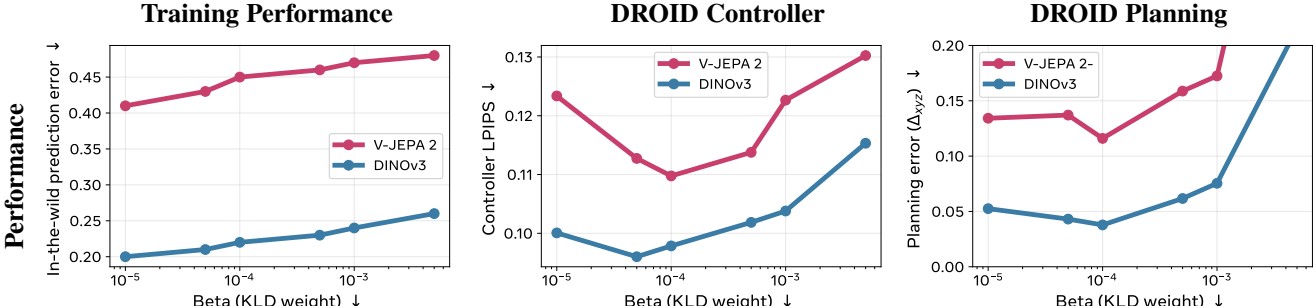

*Figure S2.* **Performance with different encoders.** We investigate the choice of encoder when training our latent action world model. We find similar trends in training performance, albeit at different absolute scales due to the differences in latent space (left). Both when training a controller and using it for planning, we find similar U-shaped curves for both encoders, with similar minima (middle and right).

As can be seen in Figure S2, both when using V-JEPA 2 or DINOv3 as our encoder, we obtain similar trends. In terms of in-the-wild prediction error, DINOv3 achieves lower values which should not be interpreted as better performance, but simply as differences in representation space structure. Turning ourselves to downstream tasks on DROID, we find that DINOV3 follows very similar trends to V-JEPA 2, achieving optimal performance with similar regularization coefficient. The performance is however significantly better for DINOv3 here. These results suggest that our findings are not specific to V-JEPA 2, but are general no matter the choice of encoder. The fact that similar regularization coefficients are optimal further reinforces the generality and applicability of our findings.

# E. Understanding Variance-Covariance-Mean regularization

In this section, we clarify the motivation behind our sparsity regularization design, specifically the $L_2$ threshold term and the Variance, Covariance, and Mean (VCM) components. The $L_1$ and $L_2$ terms serve as the foundation for sparsity. Indeed, minimizing the $L_1$ penalty can inadvertently be achieved by general norm reduction, which makes the $L_2$ constraint necessary to ensure the regularization remains effective. However, these two components alone are not sufficient, and other failure modes are still possible.

The VCM components are thus designed to help learn a well-behaved latent action space, removing potential failure modes:

- **Variance:** Without this term, the network tends to learn degenerate latents such as a unique sparse vector for all samples, e.g. $(0, 0, 1, 0, \ldots, 0)$, plus a small norm noise vector $\epsilon$ specific to each sample. This leads to latent actions that appear sparse while hiding high-dimensional information within $\epsilon$.

- **Covariance:** Without this term, non-zero latent actions dimensions become highly correlated, meaning that the intended regularization strength is circumvented.

- **Mean:** We observed that a few dimensions would adopt a high mean simply to inflate the $L_2$ norm. This learned centering prevents such behavior and stabilizes the latent action space.

The union of the base sparsity constraints ($L_1 + L_2$) and the VCM terms leads to a highly effective overall regularization strategy. To demonstrate the necessity of each component in the VCM penalty, we perform an ablation study by incrementally adding the Variance, Covariance, and Mean terms in Table S1. For these experiments, the $L_1$ coefficient is held constant at 0.05.

As shown in Table S1, each introduced component improves the raw performance of the model. By constraining the variance, covariance, and mean, the network achieves a more proper regularization even when targeting the exact same degree of sparsity.

*Table S1.* **Ablation study on the components of the VCM regularization.** The $L_1$ coefficient is fixed at 0.05. Each added component contributes to a more properly regularized space, improving overall performance metrics. We report in-the-wild prediction error, controller quality on DROID, and planning performance on DROID.

| VCM Penalties | | | Evaluation Metrics | | |
|---|---|---|---|---|---|
| Var. | Cov. | Mean | Prediction Error | Controller (LPIPS) | Planning ($\Delta_{xyz}$) |
| – | – | – | 0.63 | 0.36 | 0.49 |
| 0.1 | – | – | 0.45 | 0.13 | 0.30 |
| 0.1 | 0.001 | – | 0.44 | 0.12 | 0.15 |
| 0.1 | 0.001 | 0.1 | 0.43 | 0.12 | 0.17 |

# F. Vector quantization analysis

To better understand why discrete latent actions fail to scale to the considered setup, we analyze pretrained models at various total codebook sizes and commitment loss weights using 115k latent actions to compute various metrics. In particular, we focus on the codebook utilization, perplexity, commitment loss value, as well as 10-NN cluster entropy. All of these metrics give us a good overview of the codebook and latent action health.

Focusing on the size of the codebook, we can see on the left of Table S2 that until $|C| = 4096$ included, the codebook is rather healthy, with high utilization and NN cluster entropy. However, scaling to a larger codebook size of $|C| = 32768$, utilization drops significantly.
To better understand what breaks at this codebook size, we ablate the commitment weight $\beta$ in the middle of Table S2. We find that no matter the commitment coefficient $\beta$, we do not get a healthy codebook usage. The utilization and perplexity remain low, and the entropy does not evolve significantly. This suggests that the poor codebook usage comes from the size of the codebook itself rather than an hyperparameter choice.
For this commitment weight ablation, we also analyze the model performance on the right of Table S2. Apart from when the commitment weight is too high ($\beta = 10$), the models achieve very similar performance, which coincides with the previous analysis.

Overall, these analyses help shed light on the failure modes of vector quantization when scaling to in-the-wild videos. While other implementations of vector quantized latent actions may help improve performance, their design is out of the scope of the current work.

*Table S2.* **Understanding the failures of VQ.** We study the behavior of the learned codebook when varying the codebook size and commitment loss weight. Larger codebooks struggle with utilization, something that higher commitment loss weights cannot fix in the present setting. Downstream performance also remains quite stable on DROID.

*(a)* Codebook Size Ablation

| Codebook $|C|$ | Util. | Perplexity | Commit. loss | 10-NN Entropy |
|---|---|---|---|---|
| 16 | 100% | 13.5 (84%) | 0.020 | 2.14 |
| 1024 | 100% | 896 (88%) | 0.021 | 2.26 |
| 4096 | 99.7% | 3605 (88%) | 0.021 | 2.25 |
| 32768 | 64.8% | 16098 (49%) | 0.034 | 1.66 |

*(b)* Commitment ($\beta$) ablation ($|C| = 32$k)

| $\beta$ | Util. | Perplexity | Commit. loss | 10-NN Entropy |
|---|---|---|---|---|
| 0.01 | 52.8% | 11289 (34%) | 1.099 | 1.32 |
| 0.1 | 62.4% | 14754 (45%) | 0.094 | 1.60 |
| 0.25* | 64.8% | 16098 (49%) | 0.034 | 1.66 |
| 0.5 | 66.3% | 16543 (50%) | 0.017 | 1.69 |
| 1.0 | 66.3% | 16709 (51%) | 0.008 | 1.69 |
| 10.0 | 67.8% | 17384 (53%) | 0.0001 | 1.71 |

*(c)* Performance vs. Commitment ($\beta$)

| $\beta$ | Prediction error | Controller LPIPS | Planning error ($\Delta_{xyz}$) |
|---|---|---|---|
| 0.01 | 0.47 | 0.12 | 0.13 |
| 0.1 | 0.47 | 0.12 | 0.15 |
| 0.25* | 0.47 | 0.12 | 0.14 |
| 0.5 | 0.47 | 0.11 | 0.13 |
| 1 | 0.47 | 0.12 | 0.14 |
| 10 | 0.48 | 0.12 | 0.19 |

*\*Default value*

# G. Sampling latent actions

Throughout this work, latent actions have either been used as-is for transfer experiments, or as an interface to control the learned world model with interpretable actions. Performing planning directly in latent action space is, to the best of our knowledge, an open problem that can be made worse depending on the geometry of the latent action space.

Latent action sampling is the first process to elucidate, which varies based on the choice of latent action regularization. For **discrete latents**, the task is straightforward: sample from the codebook, possibly only for used codes. For **noisy, VAE-like latents**, the prior distribution $\mathcal{N}(0, 1)$ can be used. However, the strength of the regularization used during training will alter how closely this prior is matched, leading to suboptimal coverage of the latent action distribution. **Sparse latents** are perhaps the most challenging sampling-wise. Due to the definition of the latent action space being based on using an energy function, we have to resort to MCMC sampling techniques for EBMs (LeCun et al., 2006). A common approach is to leverage our knowledge of the energy function's gradient and use a sampler based on Stochastic Gradient Langevin Dynamics (SGLD) (Grathwohl et al., 2020; Welling & Teh, 2011). The sampling can be defined:

$$z_0 \sim p(z), \quad z_{t+1} = z_t - \frac{\alpha}{2} \frac{\partial E(z_i)}{\partial z_i} + \epsilon, \quad \text{with} \quad \epsilon \sim \mathcal{N}(0, \alpha). \tag{4}$$

Here $p$ can be a uniform distribution over the latent action space, or a Gaussian distribution for example. Similarly to using the prior distribution for noisy latents, when training a LAM we are not necessarily minimizing properly the energy function associated to our latents, which can lead to a misalignment between sampled latents and the ones inferred in practice.

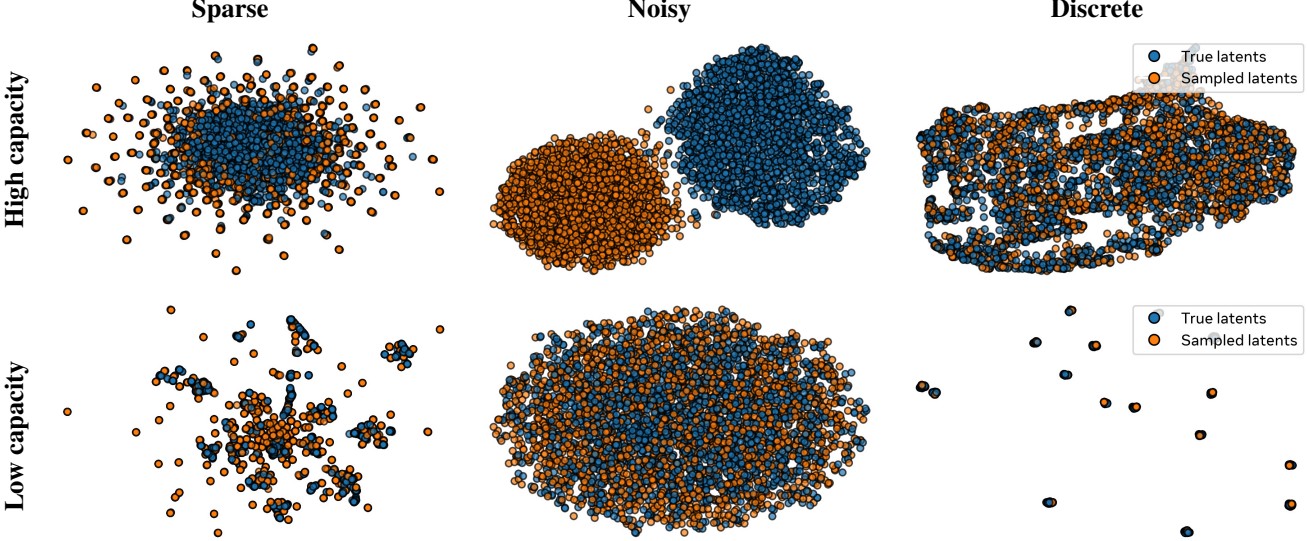

*Figure S3.* **Sampling latent actions.** For each class of latent actions and various capacities, we infer latent actions on natural videos and sample the same amount randomly. Looking at 2D visualizations obtained with UMAP (McInnes et al., 2018), we can see that high capacity latents (i.e. less constrained ones) are harder to sample as they are further away from the intended regularization or prior distribution. As the capacity gets lower, the visible overlap between sampled and true latents suggests that the sampling procedure works closer to intended.

As we can see in figure S3, the aforementioned sampling strategies are able to sample similar latents to real ones when they have a low capacity. In that case, the models were trained with stronger constraints on the latent actions which can explain why the sampling is adequate. However when the latents are less constrained, and thus have a higher capacity, the true and sampled latents are easily separable which suggests a poor sampling.

While this analysis is purely qualitative, it effectively demonstrates how sampling approaches start to break down when handling continuous latents. An interesting angle of attack to tackle this sampling problem could be to use learning based methods that make fewer assumptions about the latent action distribution, such as diffusion models (Sohl-Dickstein et al., 2015).

# H. Detailed planning results

*Table S3.* **Results on DROID.** We first train a controller to map actions to latent actions and measure the quality of the rollouts compared to the IDM (left). We then select unseen videos and infer actions based on a goal image. We measure performance as the distance to the goal (right).

| Latents | Capacity | IDM | Controller |
|---|---|---|---|
| | Low | 0.12 | 0.14 (×1.17) |
| Sparse | Mid | 0.10 | 0.12 (×1.20) |
| | High | 0.09 | 0.14 (×1.46) |
| | Low | 0.13 | 0.13 (×1.00) |
| Noisy | Mid | 0.10 | 0.11 (×1.10) |
| | High | 0.09 | 0.12 (×1.27) |
| Discrete | Low | 0.13 | 0.13 (×1.00) |
| | High | 0.11 | 0.12 (×1.02) |

| Latents | Capacity | $\Delta xyz$ (m) |
|---|---|---|
| | Low | 0.33 |
| Sparse | Mid | 0.18 |
| | High | 0.13 |
| | Low | 0.49 |
| Noisy | Mid | 0.11 |
| | High | 0.10 |
| Discrete | Low | 0.18 |
| | High | 0.14 |
| V-JEPA 2-AC | N/A | 0.15 |
| V-JEPA 2 + WM | N/A | 0.05 |

*Table S4.* **Results on RECON.** We first train a controller to map actions to latent actions and measure the quality of the rollouts compared to the IDM (left). We then select unseen videos and infer actions based on a goal image. We measure performance as ATE and RPE (right).

| Latents | Capacity | IDM | Controller |
|---|---|---|---|
| | Low | 0.23 | 0.25 (×1.11) |
| Sparse | Mid | 0.19 | 0.23 (×1.16) |
| | High | 0.17 | 0.26 (×1.51) |
| | Low | 0.24 | 0.24 (×0.99) |
| Noisy | Mid | 0.17 | 0.21 (×1.23) |
| | High | 0.17 | 0.22 (×1.29) |
| Discrete | Low | 0.24 | 0.24 (×1.00) |
| | High | 0.20 | 0.21 (×1.06) |

| Latents | Capacity | ATE | RPE |
|---|---|---|---|
| | Low | 1.68 | 0.48 |
| Sparse | Mid | 1.45 | 0.41 |
| | High | 1.43 | 0.42 |
| | Low | 2.06 | 0.55 |
| Noisy | Mid | 1.49 | 0.41 |
| | High | 1.40 | 0.40 |
| Discrete | Low | 1.81 | 0.51 |
| | High | 1.48 | 0.42 |
| NoMaD | N/A | 1.93 | 0.52 |
| NWM | N/A | 1.13 | 0.35 |

# I. Robot manipulation vs in-the-wild videos

In this section, we investigate how pretraining on DROID (Khazatsky et al., 2024) affects performance, both on qualitative examples and on planning performance.

**Qualitative analysis.** We start by comparing a model trained on YoutubeTemporal-1B with one trained solely on DROID using sparse latents with $\lambda_{l1} = 0.01$. Looking at qualitative results in Figure S4 on natural videos, we can see that a model trained exclusively on DROID struggles to model actions present in in-the-wild videos. This is even true in this scenario where we are using the inverse dynamics model, which thus represents an ideal upper bound of capabilities. Interestingly, when the action corresponds to a person entering the room, we find that the model trained on DROID makes a robotic arm appear, as it is the only moving object seen during training. While this model struggles to open and close a hand, it is however capable of animating objects that are not seen during training, such as a human walking in the scene. Looking closely we can see that the exact leg movement is not captured well, but the overall translation movement is.

These results suggest that pretraining on a more diverse dataset is beneficial to capture more diverse actions, but that even when training on a more constrained dataset, actions that still generalize can be learned. This further supports the illustration in Figure 1.

**Planning performance.** While we have previously seen that we are able to achieve good planning performance by pretraining only on in-the-wild videos, one can wonder how much the addition of domain specific data influence performance. For this, we pretrain models with a mix of DROID and YoutubeTemporal-1B data, varying the weights of the dataset between 0 and 100%.

*Table S5.* **Effect of varying DROID pretraining weight on planning.** Adding in domain data helps both the quality of rollouts and planning performance. Even a minor amount of data can yield a strong boost in performance.

| Model | DROID weight | 0% | 10% | 25% | 50% | 75% | 90% | 100% |
|---|---|---|---|---|---|---|---|---|
| Sparse | Controller LPIPS | 0.14 | 0.14 | 0.12 | 0.11 | 0.10 | 0.10 | 0.10 |
| | $\Delta$ xyz | 0.14 | 0.13 | 0.14 | 0.09 | 0.09 | 0.08 | 0.08 |
| Noisy | Controller LPIPS | 0.11 | 0.10 | 0.10 | 0.10 | 0.10 | 0.10 | 0.9 |
| | $\Delta$ xyz | 0.14 | 0.09 | 0.09 | 0.09 | 0.06 | 0.06 | 0.07 |

As we can see in Table S5, adding domain specific data can drastically help performance, even with as low as 10% in some settings. What is also interesting for our latent action model setup is that by training a latent action model with domain specific data, we can achieve very similar planning performance compared to a world model trained on the same data with access to action labels (0.06 vs 0.05 for the best model from Terver et al. (2026)). Beyond our work, these results suggest that training a latent action model on the widest range of data possible may be optimal for a diverse set of applications.

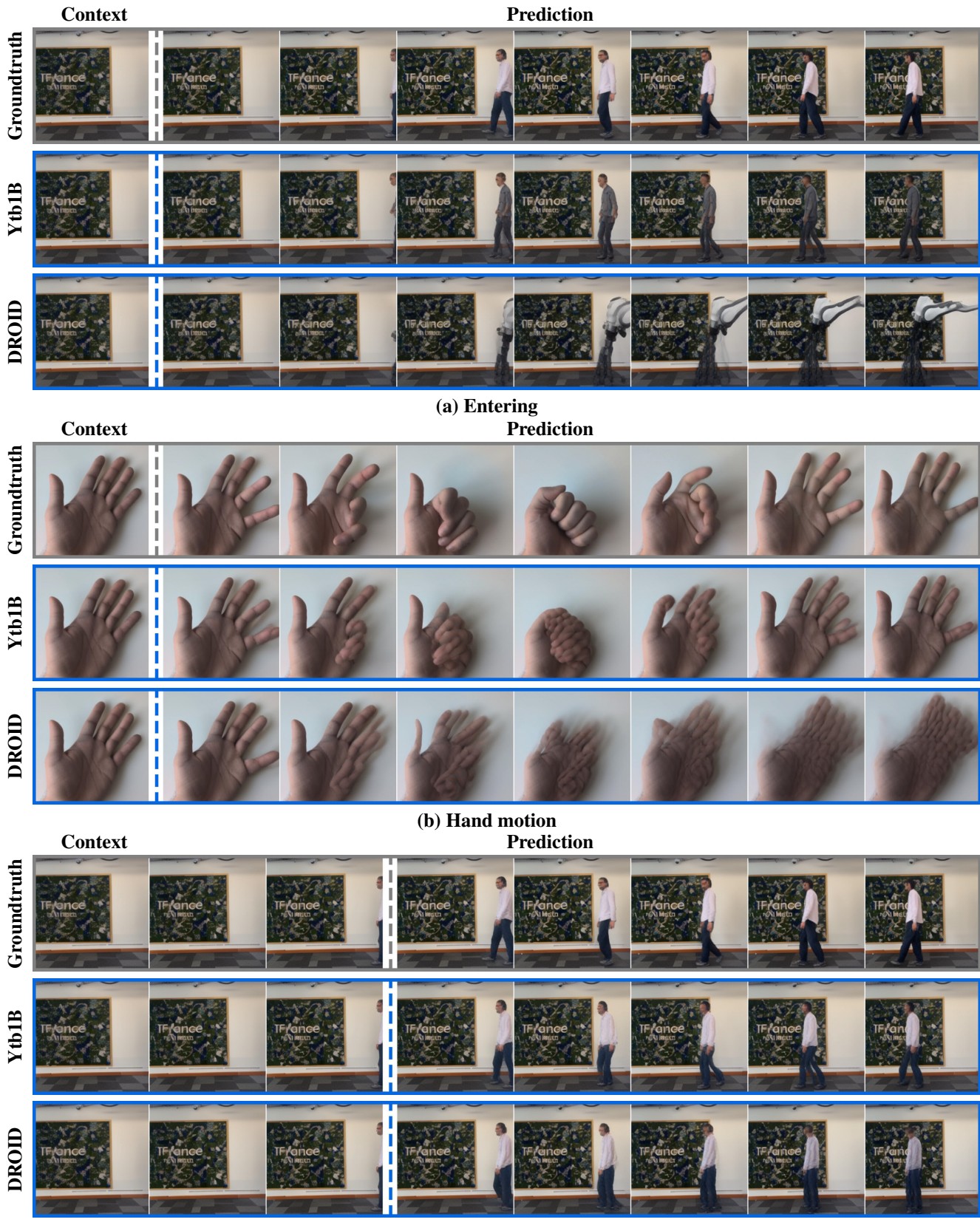

(a) Entering

(b) Hand motion

(c) Object translation

*Figure S4.* **Sample predictions using the IDM across data sources.** Top: a person entering the scene; middle: hand motion; bottom: object translation. The model trained on DROID struggles on human-centric actions outside its training distribution (entering, hand), while both models can handle simple object translation.

## J. Qualitative Impact of regularization strength

While we previously quantified the impact of latent action capacity, equivalently regularization strength, we now turn ourselves to more qualitative analyses. Throughout this section we consider noisy latents, but similar conclusions hold across regularization families.

As we can see in Figure S5, when latent actions are overly constrained, the model is unable to make a human appear. As the constraint gets weaker, we start to see the person appearing, albeit with suboptimal appearance and motion. Continuing to weaken this regularization, we start to see a better outline of the person, and a higher fidelity in motion, especially for the leg movements.

In Figure S6 we study the impact of the regularization strength when transferring movements from a human to a ball. We can see that with a too strong regularization, the ball simply continues its trajectory. We essentially have a deterministic world model. As the regularization increases, the ball slows down more until it perfectly follows the transferred motion. We then see it going perfectly left, in a straight line. This highlights the importance of adequate capacity to be able to identify interpretable actions.

While so far more capacity has been beneficial, we get a better understanding of what happens at lower constraints in Figure S7. Here we see that, although initially capacity improves the cycle consistency of actions, in some cases at higher capacity the motion is not applied to the whole human when re-inferred. This suggests a greater spatial localization of actions at higher capacity. We obtain more "precise" actions, at the cost of generality. This mirrors what is observed in planning evaluations, where the optimal latent actions spaces strike a balance between capacity and generality.

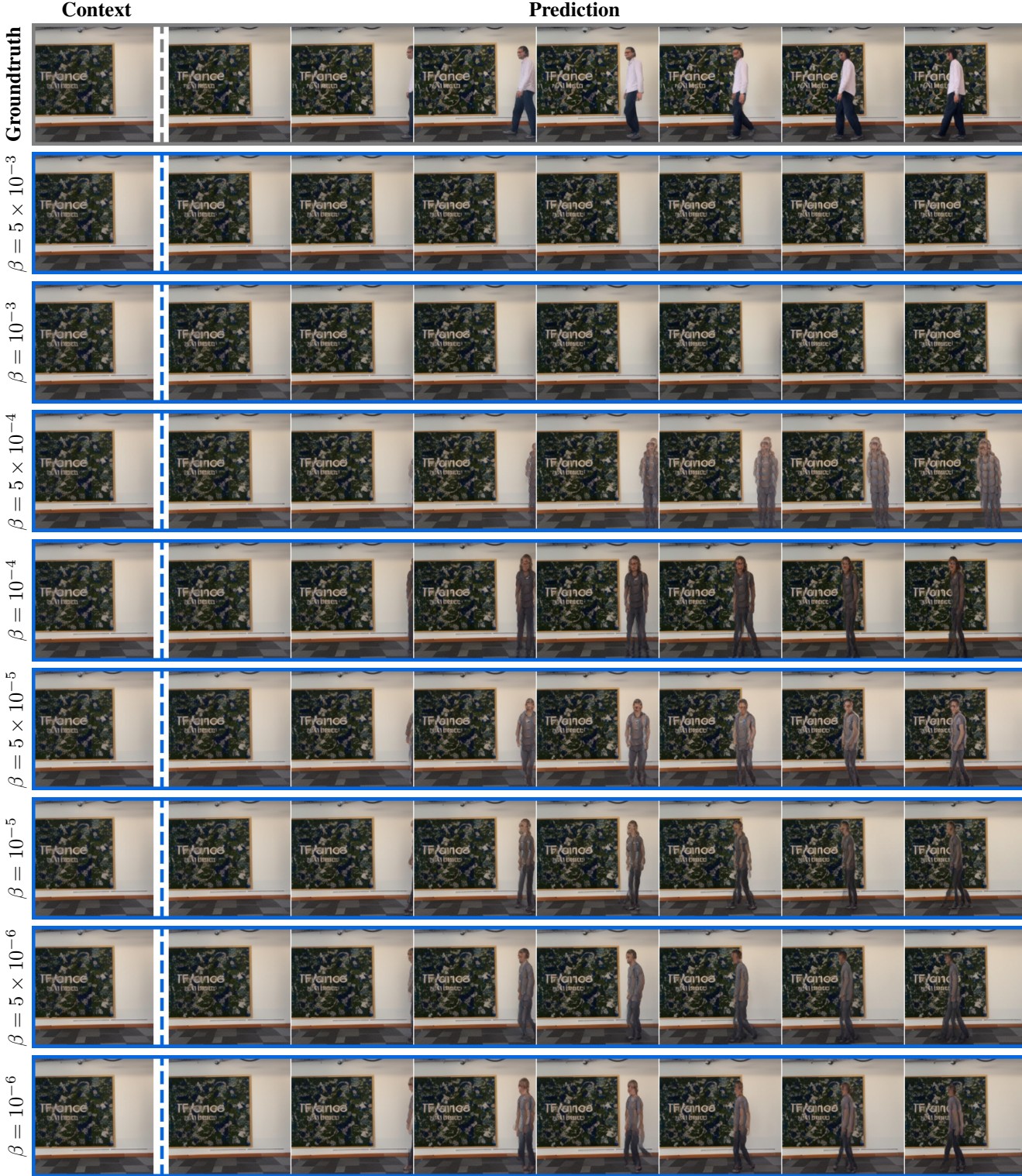

*Figure S5.* **Quality of the IDM across regularizations.** Overly constrained latents are not able to capture a person entering the room. As the capacity of the latent actions grows, both the quality of the person and leg movement increases, but plateaus after a certain point.

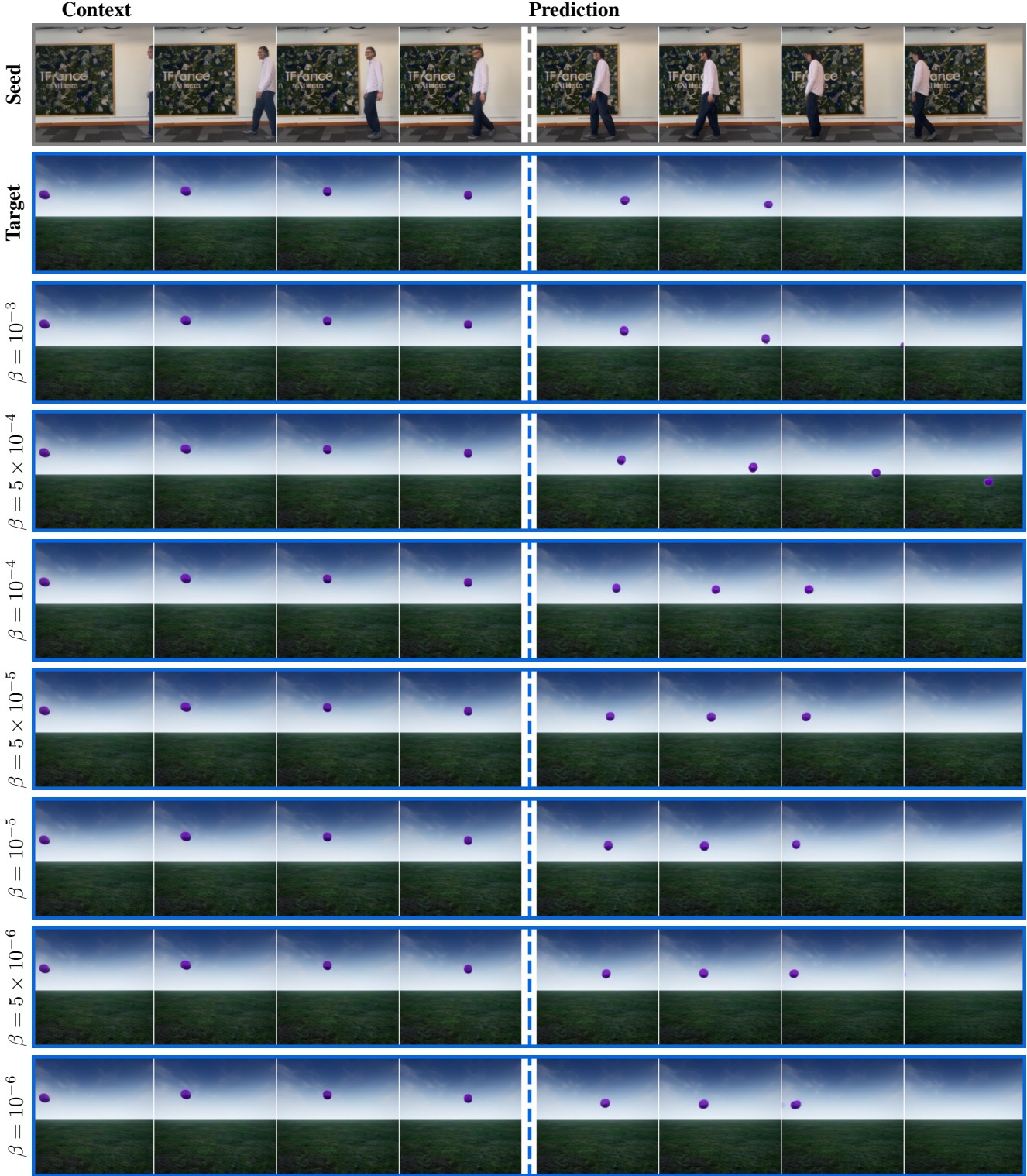

*Figure S6.* **Cross object action transfer across regularizations.** The quality of motion transfer increases with the capacity of the latents. More constrained latents either have no effect, or a weaker one.

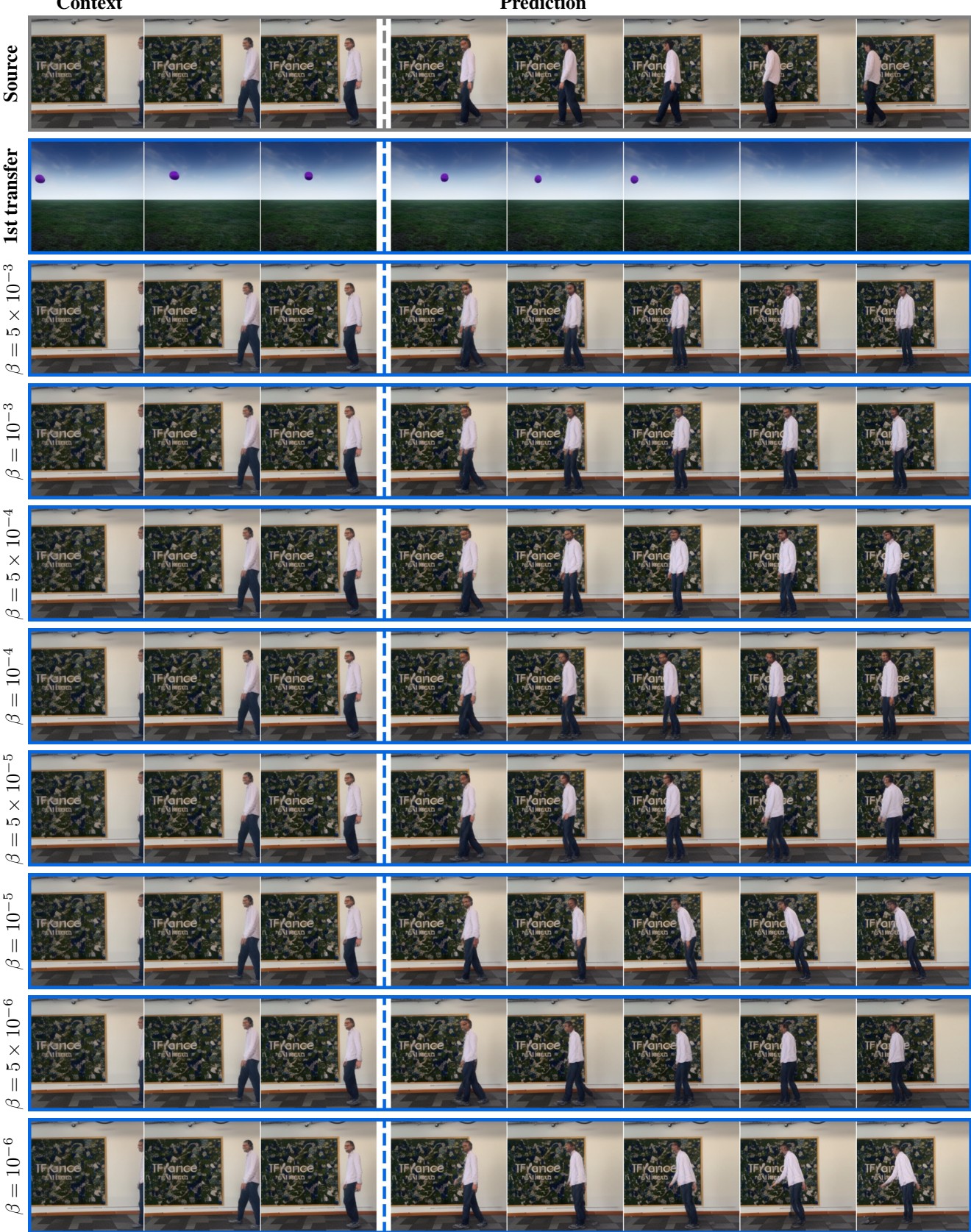

*Figure S7.* **Cycle consistency for different regularizations.** As the latent action capacity increases we obtain improved transfer. After a certain point, the movement becomes more localized and only the upper body motion is captured back.

## K. Additional IDM rollouts.

In this section we take a look at more qualitative examples of rollouts performed with the inverse dynamics models. This allows us to establish an upper bound of the performance attainable by a given model, with the caveat that models may use shortcut solutions. Similar to figure 3, we take a look at the least constrained latents for all regularizations. We focus on videos from SSv2 (Goyal et al., 2017) as a natural video dataset that are not seen during training.

As we can see in figures S8 and S9, latent actions constrained via noise addition or sparsity are able to capture the actions happening in videos, but vector quantized ones struggle more. The latter is still able to capture rough motion, but struggles with more precise ones such as the rotation of the object at the top of figure S9. Overall all of these samples corroborate our previous findings and demonstrate the usefulness of continuous regularized latent actions.

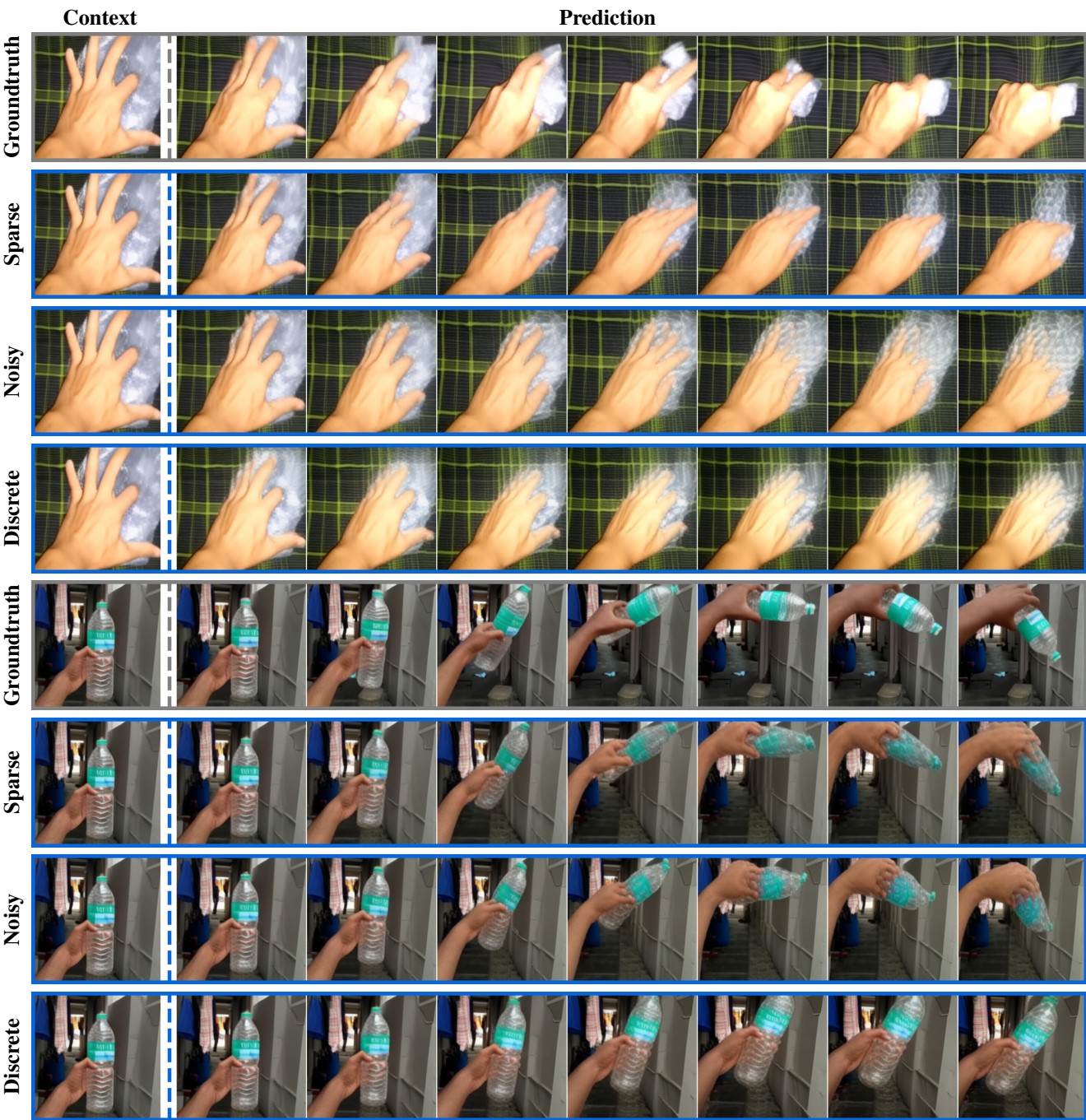

*Figure S8.* **Sample predictions using the IDM.** We illustrate the highest quality rollouts obtained with different regularization on SSv2, using the inverse dynamics model.

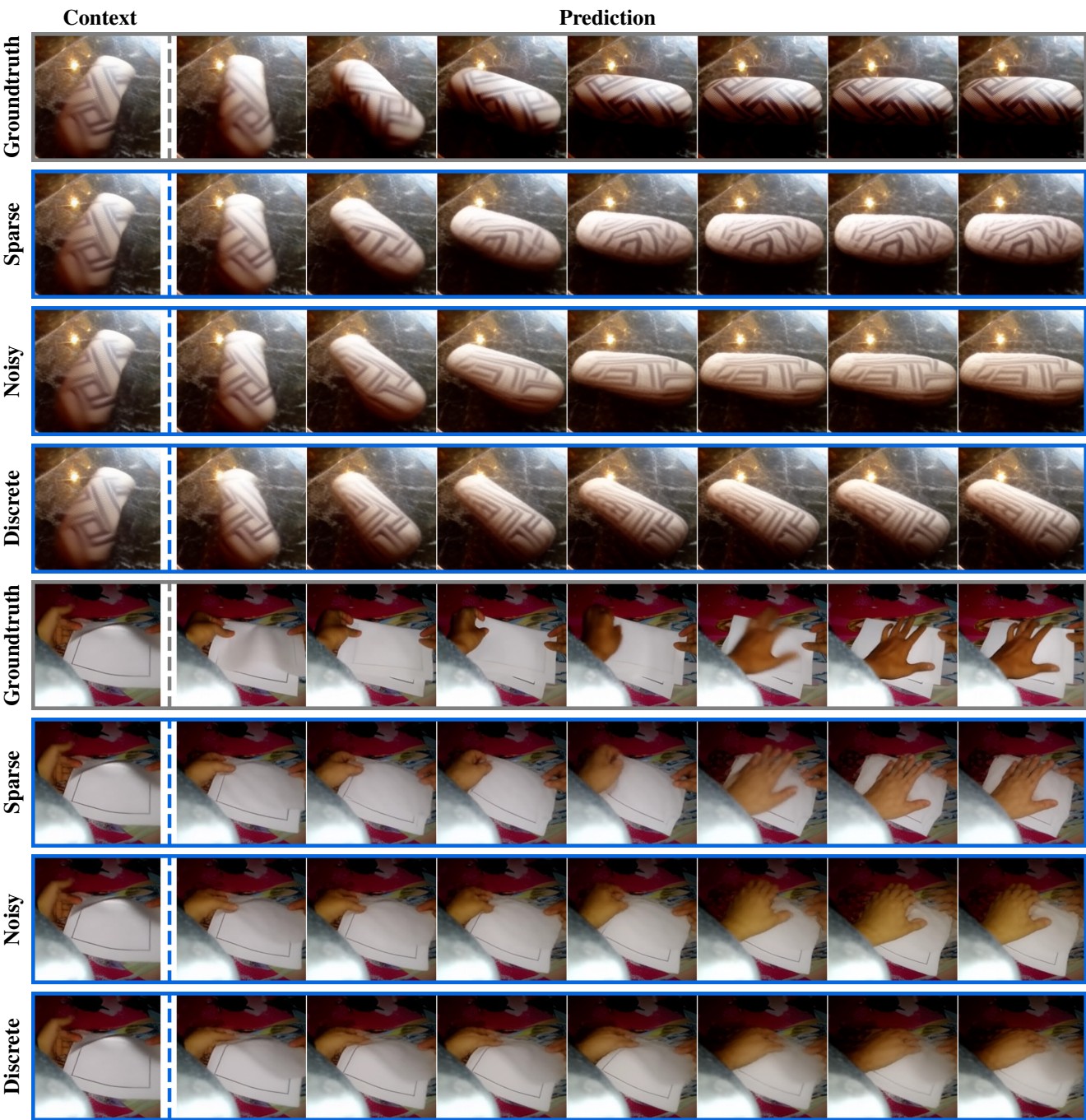

*Figure S9.* **Sample predictions using the IDM.** We illustrate the highest quality rollouts obtained with different regularization on SSv2, using the inverse dynamics model.

## L. Additional human action transfer results.

In this section, we take a look at more action transfer across scenarios. For this we consider different levels and families of regularization. We investigate four scenarios of action transfer: making someone appear and walk in a scene with someone present, two people raising their arms transferred to one person, someone entering the scene with someone else being static, someone walking in a scene. Figure S10 considers noisy latents with low capacity, Figure S11 noisy latents with high capacity latents, and Figure S12 sparse latents with high capacity. This last example has the overall highest capacity, as previously measured by prediction error.

We find that the action of someone entering an empty room is appropriately transferred, but with different behavior based on capacity. With low capacity, the newly introduced person and the one already present both start moving. At higher capacities, we see that the already present person either moves with the new character once they overlap, or disappears. We however find that if the original video contained a person standing still (third pair of row), then the person in the target video also remains still. This difference in behavior suggests that the model can distinguish humans from the background, and the latent actions affect them differently, which is a desired behavior. This is consistent with Supplementary Figure S1 where we see that the latent actions consider humans with higher priority than the background.

When transferring the motion of two people raising their right arm to a single one, we see that both arms become raised. The arms also follow the same movement as in the original video, in spite of the ambiguity of this transfer task. The arms however do not expand horizontally as much as in the original video, which we hypothesize is due to the locality of the action. This appears consistent across capacities.

Finally, when making a still person walk to the left of the scene, all capacities create movement, but at higher capacity we can see the person turn and move, which is more natural than the translation observed at lower capacity. The person only starts this motion once the motion is performed at their current location, further reinforcing the previously discussed locality.

Another positive result from these qualitative examples is that there is no leakage from the background in any video, suggesting again that models are not cheating by copying the future but learning valid latent actions.

Overall we see that actions can be adequately transferred across videos, where the difficulty of defining a clear embodiment of in-the-wild videos becomes a strength in ambiguous settings such as going from two to one person.

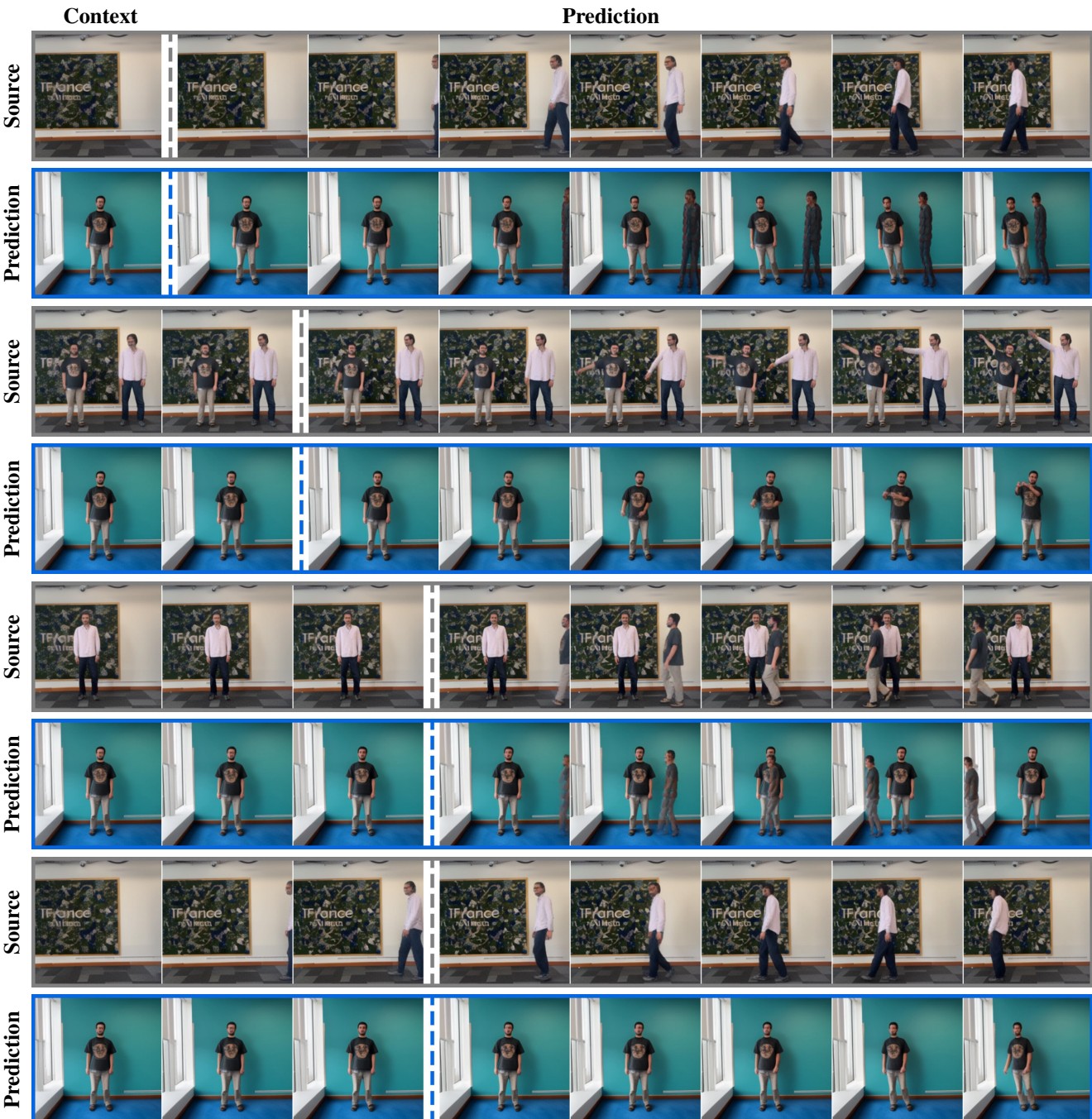

*Figure S10.* **Additional transfer results, noisy latents with** $\beta = 10^{-4}$**.** First pair of rows, making someone enter a frame with someone in it. Second pair of rows, transferring movements from two to one person. Third pair of rows, someone enters the frames with a still person in common. Fourth pair of rows, animating someone already present in the room.

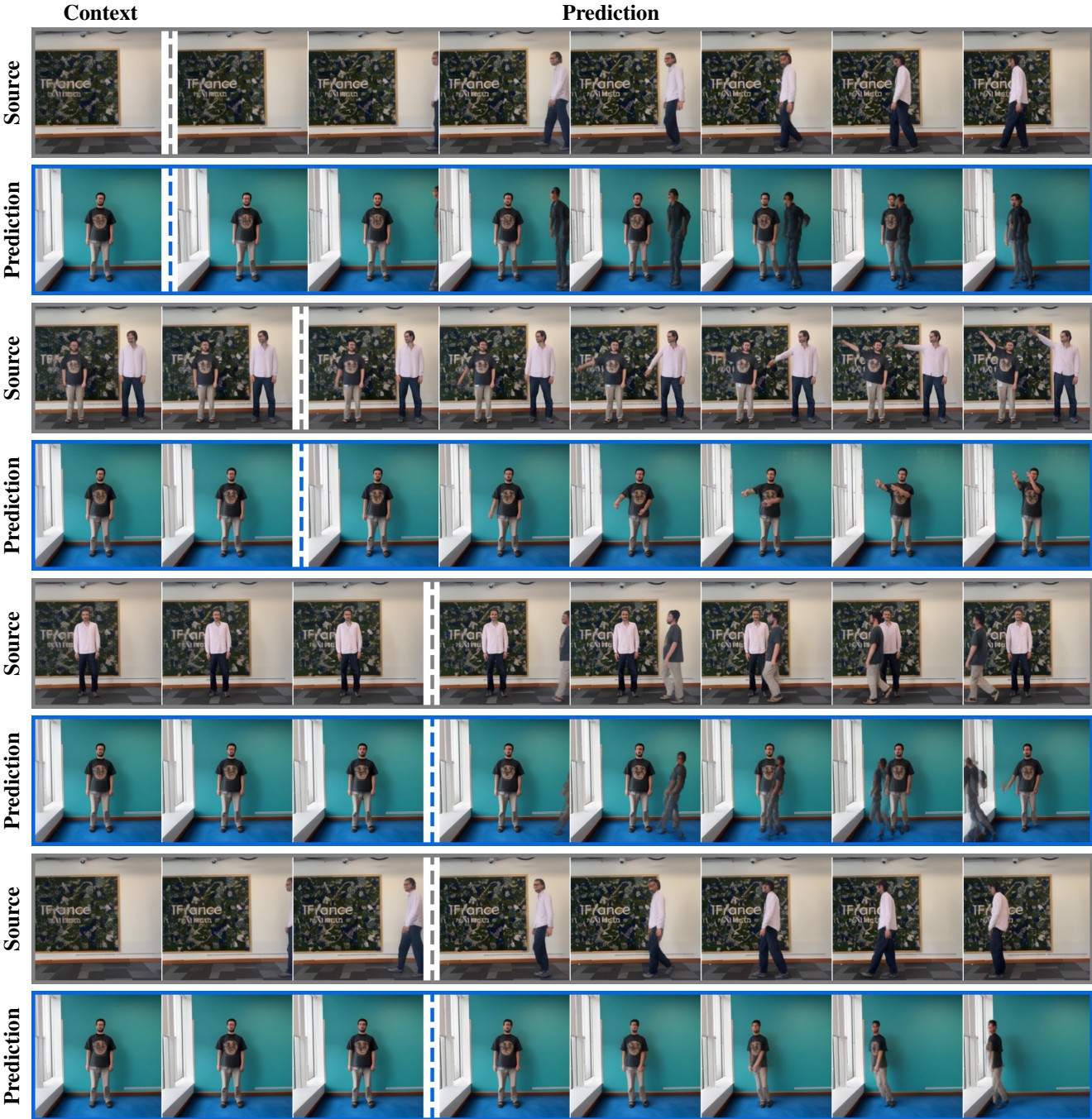

*Figure S11.* **Additional transfer results, noisy latents with** $\beta = 10^{-6}$**.** First pair of rows, making someone enter a frame with someone in it. Second pair of rows, transferring movements from two to one person. Third pair of rows, someone enter the frames with a still person in common. Fourth pair of rows, animating someone already present in the room.

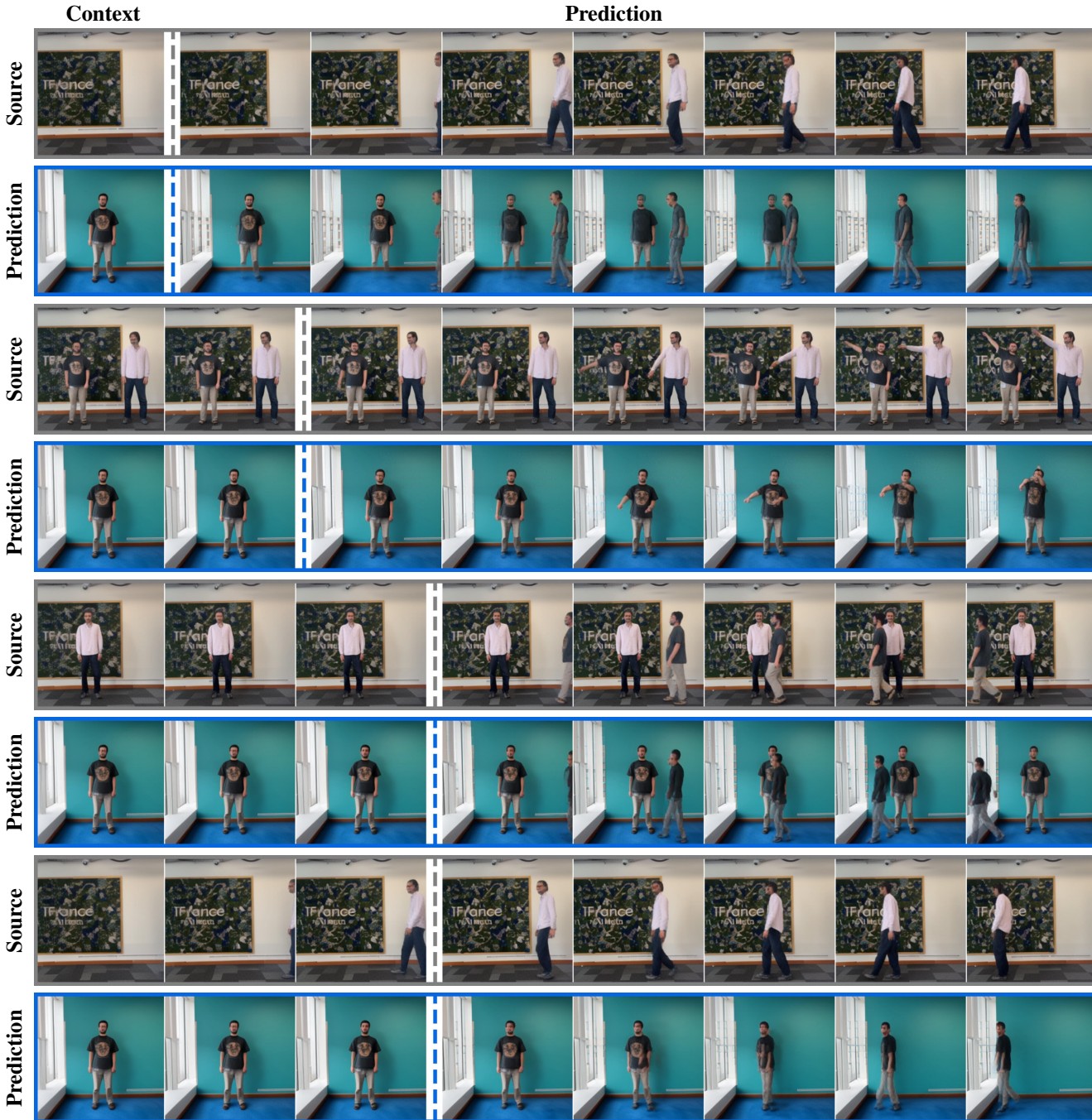

*Figure S12.* **Additional transfer results, sparse latents with** $\lambda_{l1} = 0.01$**.** First pair of rows, making someone enter a frame with someone in it. Second pair of rows, transferring movements from two to one person. Third pair of rows, someone enter the frames with a still person in common. Fourth pair of rows, animating someone already present in the room.

# M. Qualitative performance of the controllers

In this section, we take a look at rollouts produced by our learned controllers, to help understand behaviors observed in practice.

We first take a look at random samples from the validation set of RECON and DROID, using our model with the lowest LPIPS value. As we can see in Figure S13 the model is able to accurately model movements from the camera wearer, with a few caveats. In the first video, we can see that the tree is not accurately predicted once it enters the frame. This can be explained by the missing information from the beginning of the video and the model is only able to guess that the tree continues. In the second row, as the sun becomes occluded, the image gets darker. In the prediction of our model, we can see that the brightness remains high and the sun remains present in the corner of the frame, moving along with the camera. Nonetheless, we are able to accurately control the latent action world model using human interpretable actions.

On DROID in Figure S14 the model is again able to perform similar movements to the groundtruth but it struggles with making the robotic arm enter the frame. On the last row, we can see that no matter the action, nothing happens as the model did not see the arm in the video. This is a sensible failure mode. On the first row, we do see a movement of the visible part of the arm (mainly the gripper), but the rest of the arm does not appear. This again stems from a lack of information, combined with an unfamiliarity with the objects present in this video during training.

To further illustrate why the controller needs access to the representations, beyond previous intuition, we show some rollouts performed using a representation-less controller in Figure S15. Due to the different cameras possible for the videos, as well as our camera-relative latents we find that the model is not able to successfully control the robotic arm. Instead, the arm remains static. This further demonstrates the importance of representations from the past in the contextualization of latent actions.

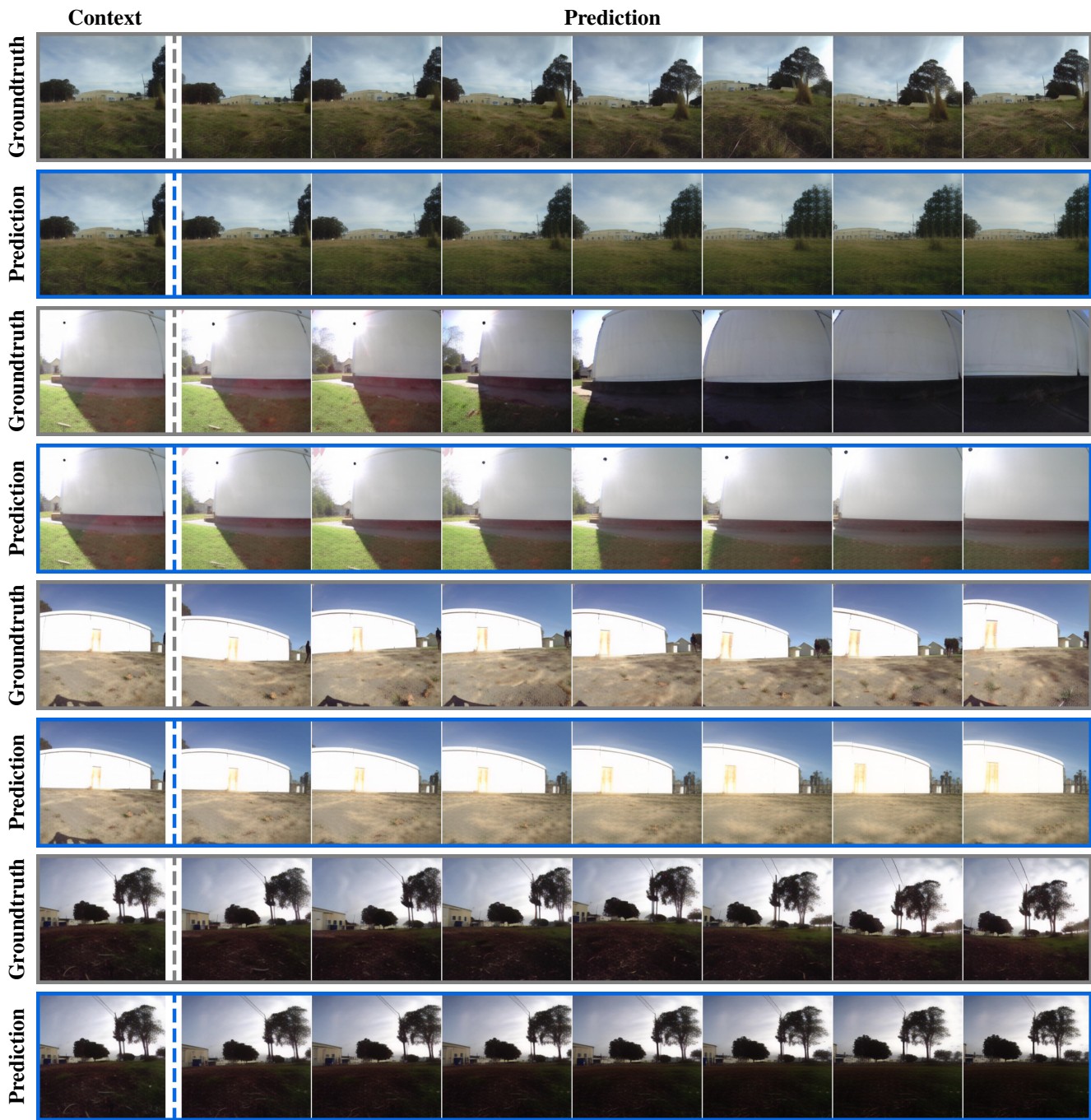

*Figure S13.* **Rollouts from the controller on RECON.** The controller can adequately map real actions to latent actions, allowing precise control of the world model.

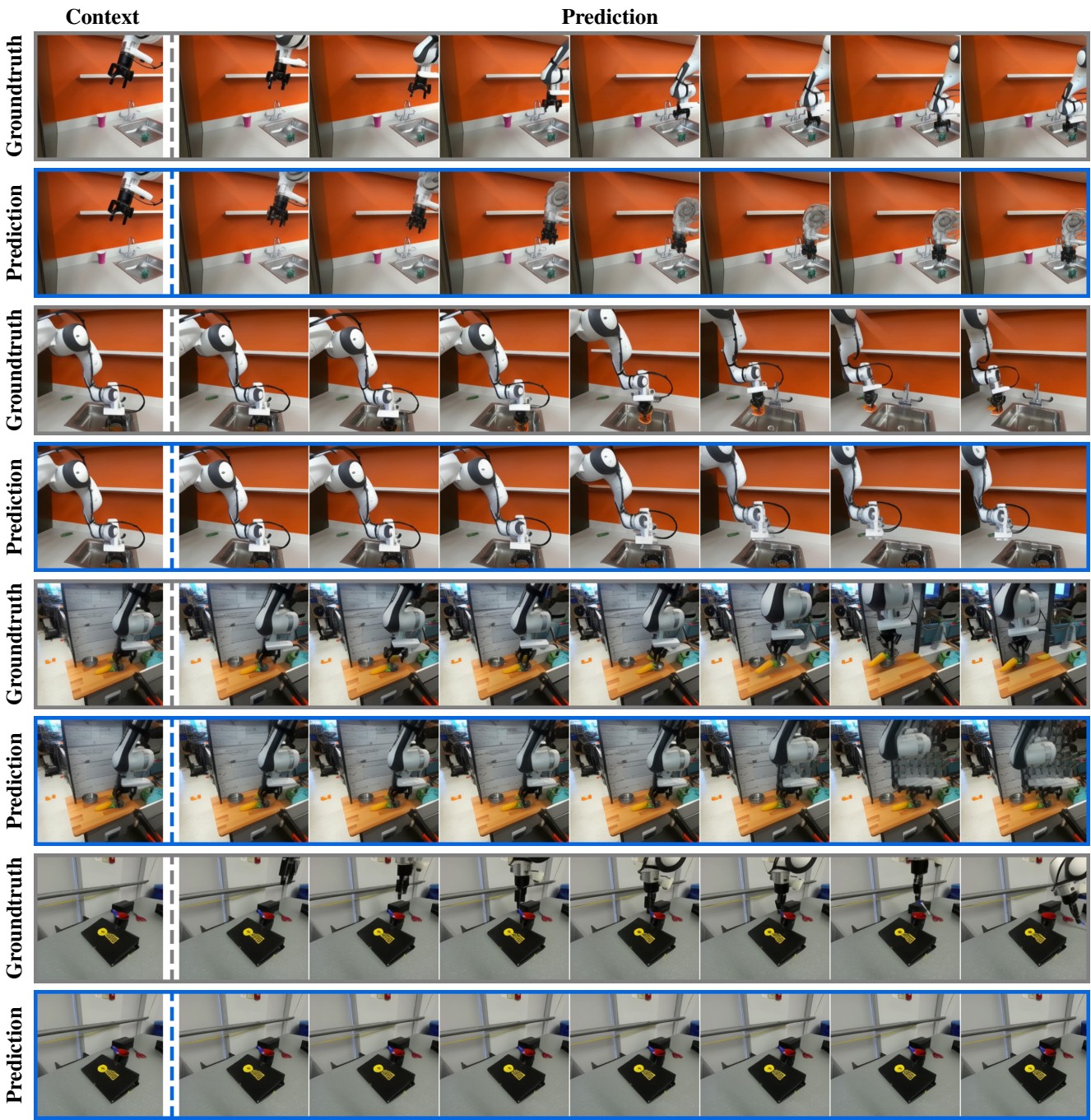

*Figure S14.* **Rollouts from the controller on DROID.** The controller can adequately map real actions to latent actions, allowing precise control of the world model when the robotic arm is in frame. When out of frame, actions become ill-defined and the model cannot make an arm appear.

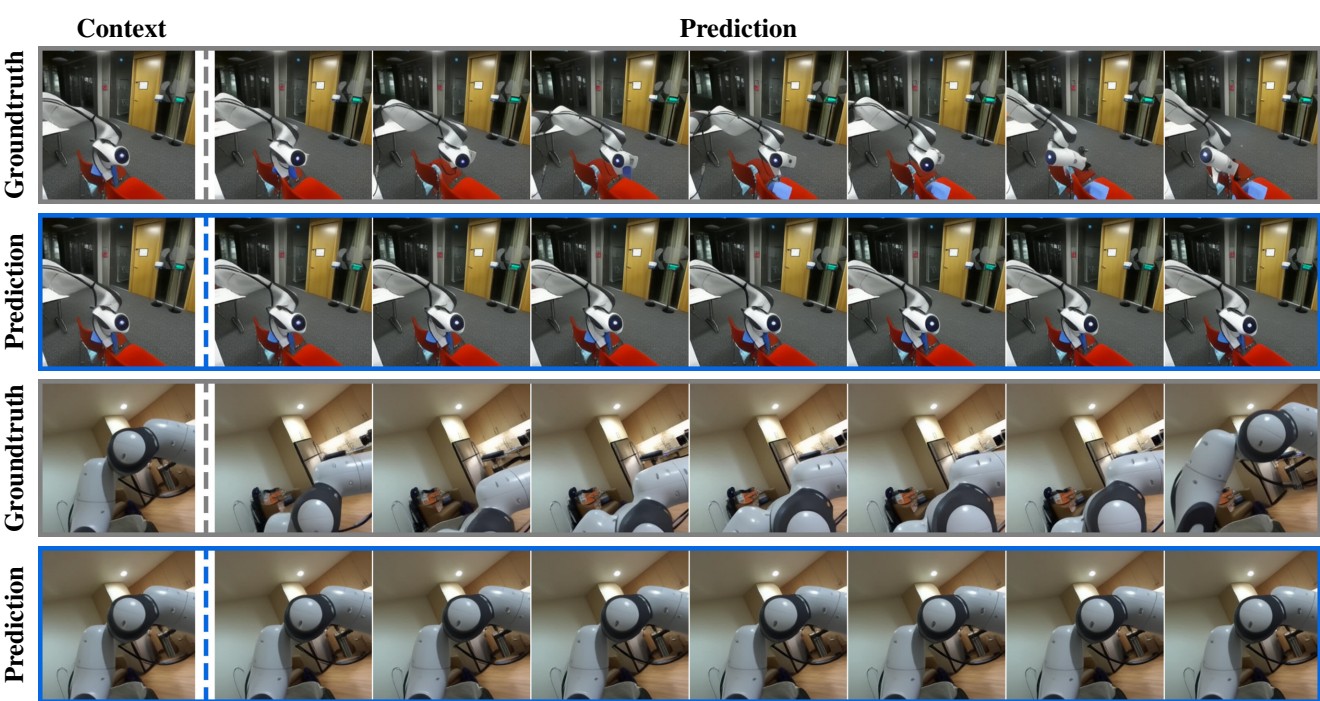

*Figure S15.* **Rollouts from the controller without representations of the past on DROID.** Due to the ambiguity of actions without knowing the position of the arm or camera, the model resorts to producing no movements.

