# OpenReview forum: "Learning Latent Action World Models in the Wild"
_ICML.cc/2026/Conference — ICML 2026 regular_

### Official Review · Reviewer_jWTp · 2026-02-21

**Soundness:** 3
**Presentation:** 3
**Significance:** 3
**Originality:** 2
**Overall Recommendation:** 5
**Confidence:** 4

**Summary:**

The paper studies the design and evaluation of latent action models that learn an action space directly from unlabeled videos in the context of world models. The main finding is that continuous latent actions (VAE/sparse) are better suited for in-the-wild general-purpose videos, as opposed to discrete latent actions that utilize a learned code book. In addition, a learned mapping between actions to latent actions is demonstrated to be useful for downstream planning tasks.

**Compliance With Llm Reviewing Policy:**

Affirmed.

**Final Justification:**

All my concerns have been addressed, I am happy to recommend accepting this paper.

**Key Questions For Authors:**

* Figure 4: I’m not sure I understand the figure. How are latent actions sampled/chosen for the prediction task? Or are they extracted from GT videos? What does it mean to “use the whole continuous space”? What does “No conditioning” and “No constraint” mean? These were not clear to me from the text or figure.
* Question: most of the latent action literature focuses on representing latent actions as holistic global vectors, under the assumption that there is a single agent in the scene; however, this may not apply to scenes with multiple objects/entities interacting (and ideally each agent would have their own latent action). How would the authors approach that? (this question is related to the final paragraph before Section 7)

**Limitations:**

Limitations are clearly discussed.

**Strengths And Weaknesses:**

**Strengths**:
* While I have some comments about the writing, the motivation and methodologies are conceptually clear.
* Insights: while the insights are not entirely surprising (continuous latent actions have more capacity), I found the rigorous analysis interesting and relevant to the community.
* I like the cycle-consistency evaluation protocol, I think it is an interesting paradigm to evaluate latent actions.
* Clear discussion of limitations.
* Extensive appendix.
* Overall, while the paper does not propose any new methods, its analyses are sound and relevant.

**Weaknesses**:
* It is very hard to tell the quality of the predictions from images, and since this paper deals with videos, it would have been nice to see a comparison of GT videos and predicted ones (either in supplementary material or a project webpage).
* The writing could be improved and some notations could be further clarified.
* The planning experiments are very simple and short-horizon.


**Minor**:
* Figure 2: clarify in the caption what are $f_{\theta}$, $g_{\phi}$ and $p_{\psi}$.
* Throughout the text, variables are used without definitions, e.g., Section 3, what are $N$, $E$, $D$?
* What does “Capacity” mean in Table 1 and 2? How do you choose/enforce “Higher Capacity”? Or is it defined based on the prediction error?

---

> ### Author Rebuttal · Authors · 2026-03-30
>
> Thank you for the thorough assessment of our work.
> We provide answers to the raised points below.
>
> **Weaknesses:**
> > It is very hard to tell the quality of the predictions from images [..] it would have been nice to see a comparison of GT videos and predicted ones.
>
> While the rebuttal format does not easily allow us to upload new videos, we are happy to include videos in the supplementary materials for an eventual camera ready version.
>
> > The writing could be improved and some notations could be further clarified.
>
> If you have any specific part of the paper that we can clarify beyond points further down in this rebuttal, please let us know and we will proceed with making it clearer.
>
> > The planning experiments are very simple and short-horizon.
>
> We agree, our experiments are a first step towards the potential of latent action world models. We would still like to point out that our experiments were directly taken from recent works.
> For navigation: Navigation World Models, and for manipulation from [1], using videos recorded by the original  authors. These are thus protocols that are relevant for the broad world modelling literature.
>
> **Minor:**
> > Figure 2: clarify in the caption what are f,g  and p .
>
> We will clarify that $f_\theta$ is the video encoder $g_\phi$ is the inverse dynamics model, and $p_\psi$ is the forward model/world model/predictor.
>
> > Throughout the text, variables are used without definitions, e.g., Section 3, what are N,E  ,D ?
>
> We will clarify that $N$ is the batch size, $D$ the dimension of the latent actions, and $E$ is a generic energy function, which is defined line 156-159.
>
> > What does “Capacity” mean in Table 1 and 2? How do you choose/enforce “Higher Capacity”? Or is it defined based on the prediction error?
>
> We define Capacity in the paragraph lines 247 to 253 as the prediction error computed on unseen in-the-wild videos (from YoutubeTemporal-1B). With all other design choices being equal, the prediction error can only be lowered with a less constrained (i.e. higher capacity) latent. This is consistent with a monotonous change in “capacity” when changing regularisation strengths.
> High or low capacity can then be obtained by lowering or increasing the strength of the constraint applied to latent actions. We will make sure that this clarification appears earlier in the paper for clarity.
>
> **Questions**
> > Figure 4: I’m not sure I understand the figure. How are latent actions sampled/chosen for the prediction task? Or are they extracted from GT videos? What does it mean to “use the whole continuous space”? What does “No conditioning” and “No constraint” mean? These were not clear to me from the text or figure.
>
> Here latent actions are extracted from the GT videos. As we discuss in Appendix E and Limitations section, sampling arbitrary latent actions remains a complex problem. We thus opted for the scenario that gives the best fighting chance to the latent actions.
>
> No conditioning means that we use a model with the same architecture but null actions. It is thus an upper bound on prediction error as it cannot handle the stochasticity of video prediction.
>
> No constraint means that we use continuous latent actions with no regularisation/bottleneck. They can thus store the most information possible in the setup. This should thus be close to a lower bound on prediction error.
>
> We will clarify these terms in the paper.
>
> >Most of the latent action literature focuses on representing latent actions as holistic global vectors, under the assumption that there is a single agent in the scene; however, this may not apply to scenes with multiple objects/entities interacting. How would the authors approach that?
>
> Indeed, this singular global vector is also a choice that we used. As we can see in figures S10 and S11, motion can be transferred from 2 people to one person. While this is a success for generic latent actions, if we purely want entity-centric latent actions we can definitely use better modeling.
> We see two main potential ideas:
> - **Object centric world models:** Recent works on world models [2] have made use of slot-attention like mechanisms to learn object centric models. This could be extended by having the same amount of latent actions as object slots for agent centric latent actions
> - **Latent actions as a grid:** we could have a latent action can only apply to a specific spatial location (defined by a patch position),  which would allow different latent actions to affect different parts of the video. By combining a bottleneck on the latent actions and a bottleneck on the grid (most patches should have no actions, i.e. all 0s) it may be possible to learn better latent actions when multiple objects interact.
>
> [1]Terver, Basile, et al. "What Drives Success in Physical Planning with Joint-Embedding Predictive World Models?." arXiv:2512.24497 (2025).
>
> [2] Nam, Heejeong, et al. "Causal-JEPA: Learning World Models through Object-Level Latent Interventions." arXiv:2602.11389 (2026).

---

> > ### Author Rebuttal · Reviewer_jWTp · 2026-04-01
> >
> > I thank the authors for their clarifications, I believe that integrating these in the next version would make the paper clearer. My concerns have been addressed. I am happy to keep my score.

---

> > > ### Author Response · Authors · 2026-04-03
> > >
> > > Thank you for your answer. We are glad that we could clarify your concerns.
> > > We will make sure that the changes are integrated in the revised manuscript.

---

### Official Review · Reviewer_evsd · 2026-03-13

**Soundness:** 2
**Presentation:** 3
**Significance:** 3
**Originality:** 2
**Overall Recommendation:** 4
**Confidence:** 3

**Summary:**

This paper studies latent action world models trained on large-scale in-the-wild video rather than on narrower domains such as games or robot manipulation. The method combines a standard IDM + forward model setup with several regularization strategies for the latent action space

**Compliance With Llm Reviewing Policy:**

Affirmed.

**Final Justification:**

Author clarify the architecture differences with previous work. Author also visualize different latent in the t-SNE space. These two explanation solve my concerns and I raise my score to positive.

**Key Questions For Authors:**

1. Can you list out the main difference beyond the same framework (IDM + latent bottleneck + forward model)?

2. Can the model you trained able to discriminate camera movement and object movement? Is there any controlled experiments?

**Limitations:**

Yes

**Strengths And Weaknesses:**

Strengths:
1. The paper studies a genuinely broader setting than prior LAM work, and makes a clear case that in-the-wild data changes the latent-action design space.
2. The comparison between sparse, noisy, and discrete regularization is useful, and the qualitative/quantitative evidence verified that discrete latents struggle more on this data.
3. The controller experiments show that the learned latent space is not only visually plausible but can usable for planning.



Weakness:
1. I think the paper is interesting and reasonably well executed, but I also share the concern that it is quite close to prior latent action papers in formulation, which is IDM + latent bottleneck + forward model.

2. I also have a question for LAM, how to distinguish between the camera movement and object movement, can you show some examples?

3. I feel the “universal action interface” claim is a bit overstated. The latent actions appear to be spatially localized and camera-relative rather than object-centric, which is interesting, but also suggests the learned interface is less semantically universal

---

> ### Author Rebuttal · Authors · 2026-03-30
>
> Thank you for your assessment of our work.
> We address your concerns below.
>
> **Weaknesses**
>
> > I feel the “universal action interface” claim is a bit overstated.
>
> We will appropriately tone down the claims. Through our experiments on multiple embodiments our work demonstrates that the learned latent actions can generalize to unseen embodiments, which is a more appropriate claim.
>
>
> **Questions**
> > Can you list out the main difference beyond the same framework (IDM + latent bottleneck + forward model)?
>
> While the components used in our work are not uncommon in the literature, the key differences arise in how we use them:
> - **IDM:** this is the most standard part, while the architecture may differ between works
> - **Forward Model:** It is usually a simple network, discarded after learning the latent actions. We propose to consider the forward model as a world model, that we aim at reusing beyond latent action training. For example, in VLA centric works, the forward model is never used once the latent actions are trained
> - **Latent Bottleneck:** we introduce a sparsity based regularizer on top of existing ones. However, our main contribution lies in the study of how such methods behave at scale, on in-the-wild data. Since the forward model is treated as a world model for later use, the interactions between the latent actions and forward model are more important than in previous work.
>
> Beyond these architectural aspects, our work is aimed at a broad study on data distribution and latent action design, which extends to evaluation protocols. Some of our protocols such as the scene change and cycle consistency measurement are particularly useful for training latent action models and designed to be reusable by the community, whereas our controller + planning evaluations are here to demonstrate a practical use of latent action world models.
>
> All of these changes combined lead to a demonstration that latent action world models can be trained purely on in-the-wild videos, and still be competitive on downstream tasks.
>
> > Can the model you trained able to discriminate camera movement and object movement? Is there any controlled experiments?
>
>
> We extracted latents from various sources, with different action types: Kinetics (activities), SSv2 (motion), RECON (navigation), EgoDex (fixed camera, hand motion), translated ImageNet images (artificial-camera movements). From these, we performed a t-SNE to visualize their clustering.
> The picture is visible at https://imgur.com/a/np1O4D7 .
>
> We see almost no overlaps between latents from translated images and EgoDex, which contain respectively only camera movement and only object movement. RECON, which consists only of camera motion overlaps almost perfectly with our synthetic translations, further supporting these results. Richer datasets tend to occupy more of the space overall, as they contain all types of actions.
> These results suggest that different types of actions/motion are well disentangled in the learned action space. We will include these results in a revised manuscript to help understanding.

---

> > ### Author Rebuttal · Reviewer_evsd · 2026-04-03
> >
> > Author clarify the architecture differences with previous work. Author also visualize different latent in the t-SNE space. These two explanation solve my concerns.

---

> > > ### Author Response · Authors · 2026-04-03
> > >
> > > Thank you for your answer, we are glad that our answers helped resolve your concerns.

---

### Official Review · Reviewer_2bzw · 2026-03-13

**Soundness:** 3
**Presentation:** 3
**Significance:** 3
**Originality:** 3
**Overall Recommendation:** 5
**Confidence:** 5

**Summary:**

This paper studies how to train latent action world models (LAMs) directly from large-scale in-the-wild videos without action annotations. The core contribution is a systematic comparison of three information-content regularizations for latent actions—sparsity, VAE-style noise, and vector-quantization (VQ)—within a joint inverse-dynamics/predictor training setup on frozen video representations, together with new, simple diagnostics for leakage and transfer. The authors find constrained continuous latents (sparse or noisy) outperform discrete codebooks in modeling complex, natural actions; learned latents tend to be camera-relative and spatially localized, yet can be transferred across videos, and a small controller can map robot actions to these latents to enable planning on DROID and RECON at performance comparable to action-conditioned baselines.

**Compliance With Llm Reviewing Policy:**

Affirmed.

**Final Justification:**

The rebuttal resolves my main concerns. In particular, the authors provide a much clearer discussion of the causal leakage issue, appropriately narrow the universality claim of the latent action space, and add useful analysis on the VQ baseline failure modes and encoder sensitivity. I also find the explanation and ablation of the sparsity regularization substantially more convincing, which improves confidence in the method’s stability and reproducibility. While some limitations remain, I think the key methodological and experimental questions have now been adequately addressed.

**Key Questions For Authors:**

1. VQ baseline analysis: Could you provide a deeper analysis of why the VQ baseline saturates—for example, codebook utilization during training, nearest-neighbor cluster entropy, and the effects of commitment weight or variants such as residual-VQ or multi-head codebooks? It would be helpful to understand whether the observed gap mainly arises from limited capacity, optimization difficulties, or a mismatch with state-dependent and spatially localized actions.

2. Encoder dependence: How sensitive are the results to the choice of the frozen encoder? Have you explored alternatives such as DINO-style encoders (which often perform well for manipulation tasks), or joint fine-tuning of the encoder together with LAM training?

3. Sparsity regularization design: Could you clarify the motivation behind the sparsity regularization—specifically the L2 threshold term and the sign of the mean term in VCM? Ablation studies that remove each component would help determine which parts of the regularization are truly necessary.

**Limitations:**

Yes.

**Strengths And Weaknesses:**

Strength:

1. Presents a systematic study of information-regularized latent actions in challenging in-the-wild video settings. It introduces two diagnostics—scene-cut sensitivity to detect leakage and cycle-consistency transfer to test action reusability—and reveals that learned latent actions exhibit camera-relative, spatially localized structures enabling cross-object motion transfer.

2. The method is trained at scale on YouTubeTemporal-1B with a frozen V-JEPA 2 encoder and evaluated across diverse domains including human activity, navigation, and manipulation. Experiments compare multiple regularization strategies and show that constrained continuous latents outperform VQ representations while supporting downstream planning via CEM.

3. The work addresses the lack of action labels and shared embodiment by demonstrating that latent actions learned from unlabeled videos can support planning. It also provides empirical evidence that constrained continuous latent representations scale better than discrete ones for large, diverse video datasets.

Weakness:

1. Methodological limitations
IDM-based training can still permit causal leakage; while scene-cut diagnostics help detect it, they do not fully guarantee leakage-free learning compared with explicitly causal JEPA-style approaches. In addition, the proposed sparsity regularization (VCM + L1 + L2 thresholding) is somewhat ad hoc and insufficiently justified, raising concerns about stability and reproducibility.

2. Controller and embodiment issues
The learned camera-relative latent actions enable cross-object transfer but complicate controller design. In practice, the controller collapses to a no-op without past representations, which limits the claim that the learned latent space provides a universal action interface.

3. Experimental gaps
Discrete (VQ) baselines may be under-explored, with limited analysis of alternative quantization strategies. Planning evaluations rely on short horizons and proxy metrics with few episodes, lack real-robot experiments, and trail stronger navigation baselines such as NWM.

---

> ### Author Rebuttal · Authors · 2026-03-30
>
> Thank you for the assessment of our manuscript and very relevant comments.
>
> > IDM-based training can still permit causal leakage; [...] guarantee leakage-free learning compared with causal JEPA-style approaches
>
> Any method that uses the target to find the latent action allows causal leakage, whether an IDM or through optimization. A solution would be to model $z_t \sim p(s_{0:t})$, which is an open challenge. If you have a precise method in mind, we would be happy to discuss it.
>
> > The learned camera-relative latent actions enable cross-object transfer but complicate controller design. [..]  limits the claim that the learned latent space provides a universal action interface.
>
> We agree that this locality can complicate downstream use. However, we demonstrated how to leverage them successfully for downstream tasks, partially alleviating this concern. We will tone down the universality claims.
>
> > Could you provide a deeper analysis of why the VQ baseline saturates ?
>
> We analyzed pretrained models at various total codebook size using 115k latent actions to compute metrics.
>
> | Codebook | Utilization | Perplexity | Commit. loss | 10-NN Cluster Entropy ($ln(10)=2.3$) |
> |:---|:---|:---|:---|:---|
> | 16 | 100% | 13.5 (84%) | 0.020 | 2.14 |
> | 1024 | 100% | 896 (88%) | 0.021 | 2.26 |
> | 4096 | 99.7% | 3605 (88%) | 0.021 | 2.25 |
> | 32768 | 64.8% | 16098 (49%) | 0.034 | 1.66 |
>
> Until $|C| = 4096$ included the codebook is rather healthy, with high utilisation and NN cluster entropy, but utilization drops at $|C| = 32768$.
>
> We also performed ablations on the commitment weight $\beta$ for our largest codebook (32k).
>
> | $\beta$ | Utilization | Perplexity | Commit. loss | 10-NN Cluster Entropy |
> |:---|:---|:---|:---|:---|
> | 0.01 | 52.8% | 11289 (34%) | 1.099 | 1.32 |
> | 0.1 | 62.4% | 14754 (45%) | 0.094 | 1.60 |
> | 0.25 (default) | 64.8% | 16098 (49%) | 0.034 | 1.66 |
> | 0.5 | 66.3% | 16543 (50%) | 0.017 | 1.69 |
> | 1.0 | 66.3% | 16709 (51%) | 0.008 | 1.69 |
> | 10.0 | 67.8% | 17384 (53%) | 0.0001 | 1.71 |
>
> No matter the commitment coefficient $\beta$, we do not get a healthy codebook usage. The utilization/perplexity remains low, and the entropy does not evolve significantly. This suggests that the codebook is simply too large.
>
> Looking at model performance:
>
> | $\beta$ | 10 | 1 | 0.5 | 0.25 (default) | 0.1 | 0.01 |
> |:---|:---|:---|:---|:---|:---|:---|
> | Prediction error | 0.48 | 0.47 | 0.47 | 0.47 | 0.47 | 0.47 |
> | Controller LPIPS (DROID) | 0.12 | 0.12 | 0.11 | 0.12 | 0.12 | 0.12 |
> | Planning error ($\Delta_{xyz}$) | 0.19 | 0.14 | 0.13 | 0.14 | 0.15 | 0.13 |
>
> Apart from when the commitment weight is too high, the models achieve very similar performance.
> Our work uses the UniVLA implementation, which can be described as successful for latent actions. We will make sure to mention in our work that more advanced discretization techniques could improve the behaviour.
>
> Overall, these analyses help shed light on the failure modes of VQ when scaling to in the wild videos.
>
>
> > How sensitive are the results to the choice of the frozen encoder?
>
> We repeated experiments using a DINOv3-L encoder. Due to space constraints in our answers, the detailed numbers are in our answer to Reviewer CX8J.
>
> We see similar trends as V-JEPA 2 when looking at downstream performance, with higher planning performance for DINOv3. Interestingly, the same hyperparameters are optimal across both encoders, demonstrating that our analysis can generalize.
>
> >  Sparsity regularization design: Could you clarify the motivation behind the sparsity regularization—specifically the L2 threshold term and the sign of the mean term in VCM?
>
> Each component can be motivated as:
> - **L2 thresholding**: Minimizing L1 can be done by norm reduction. Without this L2 constraint, the regularization is ineffective
> - **Variance**: Without it, the network learns a latent of the form (0,0,1,0,....0) + $\epsilon$, to be sparse but retain high-dim. information through $\epsilon$ .
> - **Covariance**: Similar issue, dimensions were very correlated, meaning that the regularization was not working as intended
> - **Mean**: A few dimensions had high mean to inflate the L2 norm, so this learned centering helps get a nicer behaved latent action space
>
> V+C+M are here to help learn a nicely behaved latent action space, while L1 + L2 are the base for sparsity. The union of both leads to an effective regularisation. These explanations will be added to the paper.
>
> > Ablation studies that remove each component (VCM) are necessary.
>
> With an L1 coefficient of 0.05:
>
> | Var. | Cov. | Mean | Prediction error | Controller error (LPIPS) | Planning error ($\Delta_{xyz}$) |
> |:---|:---|:---|:---|:---|:---|
> | | | | 0.63 | 0.36 | 0.49 |
> | 0.1 | | | 0.45 | 0.13 | 0.30 |
> | 0.1 | 0.001 | | 0.44 | 0.12 | 0.15 |
> | 0.1 | 0.001 | 0.1 | 0.43 | 0.12 | 0.17 |
>
> Each component improves raw performance, as it allows for a more proper regularization even with the same target sparsity.  These analyses will be added to the manuscript.

---

> > ### Author Rebuttal · Reviewer_2bzw · 2026-04-04
> >
> > The rebuttal resolves my main concerns. In particular, the authors provide a much clearer discussion of the causal leakage issue, appropriately narrow the universality claim of the latent action space, and add useful analysis on the VQ baseline failure modes and encoder sensitivity. I also find the explanation and ablation of the sparsity regularization substantially more convincing, which improves confidence in the method’s stability and reproducibility. While some limitations remain, I think the key methodological and experimental questions have now been adequately addressed.

---

> > > ### Author Response · Authors · 2026-04-06
> > >
> > > We are glad that our clarifications and experiments were able to address your concerns.
> > > We will make sure that everything is included in the revised manuscript.

---

### Official Review · Reviewer_CX8j · 2026-03-13

**Soundness:** 3
**Presentation:** 4
**Significance:** 3
**Originality:** 3
**Overall Recommendation:** 5
**Confidence:** 4

**Summary:**

The work addresses the problem of having magnitudes of unlabelled video data in the wild and how it can be utilized to abstract the latent factors (actions) governing the latent dynamics. The work studies different bottlenecks (noisy, sparsity, and quantized) bottlenecks and their corresponding capacities on the effect of extracting such latent action space and found that discrete bottlenecks hinder the performance. They also quantitatively evaluate generalization by transferring such actions between videos and show strong quantitative and qualitative results.

**Compliance With Llm Reviewing Policy:**

Affirmed.

**Key Questions For Authors:**

1. Why does the work not include the transfer in the LAM space instead of the pixel space?
2. The work mentioned the PCA of the LAM latents but does not support this hypothesis. Can the work include such an analysis? conditioning on the PCAs of the LAM when transferring between videos?
3. More explanation of the future leakage in  Figure S1 is needed because claims strongly rely on this.
4. Would the findings change when using a different base model? Are the findings restricted to VJEPA?

**Limitations:**

Yes.

**Strengths And Weaknesses:**

### Strengths

1. Strong generalization results using the cycle transfer.
2. Qualitative results substantiate the claims of the paper clearly.
3. Delivery and clarity are quite good with a principled study of current abstraction approaches.
4. Findings are insightful to guide the field for learning LAMs.

### Weaknesses

1. Use of the visual space metrics, like LPIPS, to quantify the results has lots visual of confounders. This can be mitigated by using the probes for the actual action space.
2. Cycle consistency results include completely OOD examples that would not show useful results. Images within the video that break the temporal consistency can have more information for such evaluation.
3. Figure S1. Future leakage challenges the Table 2 results, which the authors claim in line 314. Good transfer can be explained by future leakage in case of high capacity, not the action space abstraction.
4. Figure 5 is missing the LAM model in A.

---

> ### Author Rebuttal · Authors · 2026-03-30
>
> Thank you for the thorough assessment of our work.
>
> > Use of the visual space metrics, like LPIPS, to quantify the results has lots visual of confounders. [..] Why does the work not include the transfer in the LAM space instead of the pixel space?
>
> We reported LPIPS as it is more interpretable and encoder-agnostic. The trends using the prediction error in latent space shows the same results across experiments, albeit in a less interpretable way. We will make sure to include the same tables with both metrics in the revised manuscript.
>
> > Cycle consistency results include completely OOD examples [..] Images within the video that break the temporal consistency can have more information
>
> We tested the following protocol: infer actions from the first 16 frames of a video, then apply them to the subsequent 16 frames (4s delta). This roughly maintains semantics while breaking temporal consistency.
>
> On average across all noisy latents models, we previously had an increase in prediction error (or LPIPS) of around 25% after the cycle. With this new protocol, the increase is 20% on average. The best model only sees an increase of 9% with this new protocol.
>
> > Figure S1. Future leakage challenges the Table 2 results. Good transfer can be explained by future leakage in case of high capacity
>
> In Figure S1, we can see that the model generates a very poor reconstruction. Some components which are useful to model are transferred more properly, such as the human.
> However, in Table 1, we see the distance after a scene change to be between 0.50 and 0.69. Two random videos would be at 0.72 on average. The model thus outputs latents with little information about the next frame.
>
> This evaluation also tests the worst case of: When it is the only solution, can the model encode the next frame ? Figure S1 shows that the models can partially encode some of the next frame information, but little of it. This is reassuring, as it is not possible to prove that we cannot decode the next state from the latents in the general case.
>
> Qualitative results in Figures 6 and S6 clearly indicate that when transferring latent actions in practice, no information about the scene is kept. These results further support the general lack of leakage in natural settings.
>
> > The work mentioned the PCA of the LAM latents but does not support this hypothesis. [..] conditioning on the PCAs of the LAM when transferring between videos?
>
> Our mention of PCA of actions in videos in the introduction was a way to motivate our work. Indeed, the most common and important motions should be camera motions, hand motions, etc.
> The non-smoothness of our latent action space led us to perform a t-SNE (rather than PCA) of learned latent actions across datasets to understand the structure of the space. In the figure visible at https://imgur.com/a/np1O4D7 , we see a clear distinction between data containing only camera motion (RECON) and no camera motion (EgoDex), with less clear trends for more diverse data sources.
>
> We also computed PCs of latent actions across datasets, then used the PCs at various scales as latent actions. While all were not interpretable, looking at the top PCs we found some that represented top-down camera motion, making a person look up, or zooming out. Since the learned latent action space can be highly non linear, these analyses must be interpreted carefully.
>
> > Would the findings change when using a different base model?
>
> We repeat the experiments with the noisy latent actions using DINOv3-L as our encoder
> The results are summarized in the table below:
>
> | $\beta$ | 5e-3 | 1e-3 | 5e-4 | 1e-4 | 5e-5 | 1e-5 |
> | :--- | :--- | :--- | :--- | :--- | :--- | :--- |
> | in-the-wild prediction error|
> | VJEPA 2-L | 0.48 | 0.47 | 0.46 | 0.45 | 0.43 | 0.41 |
> | DINOv3-L | 0.26 | 0.24 | 0.23 | 0.22 | 0.21 | 0.20 |
> | DROID controller LPIPS |
> | VJEPA 2-L | 0.13 | 0.12 | 0.11 | 0.11 | 0.11 | 0.12 |
> | DINOv3-L | 0.12 | 0.11 | 0.11 | 0.11 | 0.10 | 0.11 |
> | DROID planning ($\Delta_{xyz}$)|
> | VJEPA 2-L | 0.49 | 0.17 | 0.16 | 0.11 | 0.14 | 0.13 |
> | DINOv3-L | 0.22 | 0.08 | 0.06 | 0.04 | 0.05 | 0.05 |
>
>
> We find similar trends for both encoders. First, as we relax the constraint on the latent actions, the prediction error decreases, with different scales due to the different encoders.
>
> Second, DINO achieves very similar Controller LPIPS to V-JEPA 2, with a similar trend. Performance improves as the regularisation is relaxed, but then we see a decrease in performance when it starts to be too low.
>
> Third, applying the model to planning tasks, we can see that DINOv3 outperforms V-JEPA 2, but still achieves the best performance for the same regularisation strength ! This performance gap between the two is consistent with the findings of [1].
> Overall, these results are encouraging regarding the generalisation of our results to other encoders.
>
> [1]Terver, Basile, et al. "What Drives Success in Physical Planning with Joint-Embedding Predictive World Models?." arXiv:2512.24497 (2025).

---

> > ### Author Rebuttal · Reviewer_CX8j · 2026-04-03
> >
> > Thank you for the clear and thorough rebuttal.
> >
> > The added clarifications address most of my concerns. In particular, the improved cycle-consistency protocol and the additional results with DINOv3 strengthened the paper.
> >
> > I still have one question regarding the evaluation metric. Are you using the standard VGG-based LPIPS? If so, it is not fully encoder-agnostic and may bias the evaluation. Using a more general, semantically aligned distance, such as one based on DINOv3, would likely make the results easier to interpret.
> >
> > Overall, I am satisfied that the main issues have been addressed, and I maintain my positive score.

---

> > > ### Author Response · Authors · 2026-04-03
> > >
> > > Thank you for your answer. We are glad that our answers alleviated your concerns, and we will make sure to include the new results in an eventual revision.
> > >
> > > Regarding your question, we indeed used the VGG-based LPIPS. We agree that this (or the use of any decoder based metric in general) can add a bias, which is why we in general would advise to look at trends more than raw values when comparing different encoders.
> > > Using DINOv3 also has a bias, in particular when using a DINOv3 encoder as in our new experiments, but with a higher focus on the semanticity of the representations.
> > >
> > > To provide more quantitative results on the choice of metric, we first looked at the correlation between LPIPS and the distance in normalized DINOv3 representation space. Over 1024 videos from Kinetics 400, we find a Spearman correlation of 0.60, which is coherent with the DINOv3 metric being more semantic focused.
> > >
> > > We further evaluated a series of our V-JEPA 2 based models with noisy latent actions using both metrics to see if any difference arises:
> > >
> > > | $ \beta $ | 5e-3 | 1e-3 | 5e-4 | 1e-4 | 5e-5 | 1e-5 |
> > > | :--- | :--- | :--- | :--- | :--- | :--- | :--- |
> > > | *Scene cut increase* | | | | | | |
> > > | VGG-based | x2.1 | x2.3 | x2.4 | x2.6 | x2.5 | x2.4 |
> > > | DINOv3-based | x2.5 | x2.8 | x2.9 | x3.0 | x3.0 | x2.9 |
> > > | *Cycle consistency increase* | | | | | | |
> > > | VGG-based | x1.08 | x1.16 | x1.17 | x1.19 | x1.23 | x1.20 |
> > > | DINOv3-based | x1.04 | x1.11 | x1.09 | x1.12 | x1.16 | x1.12 |
> > > | *Controller performance on DROID* | | | | | | |
> > > | VGG-based | 0.13 | 0.12 | 0.11 | 0.11 | 0.11 | 0.12 |
> > > | DINOv3-based | 0.14 | 0.13 | 0.12 | 0.12 | 0.13 | 0.13 |
> > > | *Planning performance ($\Delta_{xyz}$)* | | | | | | |
> > > |  | 0.49 | 0.17 | 0.16 | 0.11 | 0.14 | 0.13 |
> > >
> > > We can see that  the trends are extremely similar, but that the DINO metric shows more extreme increase in error around scene changes. This further suggests that most of the semantics is destroyed through this operation, while the model can do well at cycle consistency of actions.
> > >
> > >
> > > Overall, we agree that the choice of encoder (VGG/DINOv3) is an important aspect for perceptual metrics, but that a more detailed analysis would be better suited as the focus of another study. Notably, more models and human evaluations would be necessary to understand which metric is “better”, and what aspect each metric fails to capture. We will be happy to add a start to this broad discussion in the appendix of our work.

---

### Decision · Program_Chairs · 2026-04-30

**Decision:**

Accept (regular)

**Comment:**

The paper received broadly positive reviews. Reviewers found the work timely and well executed, and appreciated the systematic study of latent action bottlenecks for world models trained on large-scale in-the-wild video. In particular, the comparison between continuous and discrete latent-action bottlenecks, together with the proposed diagnostic protocols such as cycle-consistency transfer and leakage analysis, provides useful insights for the community.

The rebuttal addressed the main concerns effectively by clarifying the evaluation metrics, providing additional analysis of leakage and VQ failure modes, showing similar trends across different encoders, and better justifying the sparsity regularization. While some limitations remain, especially regarding claim calibration and downstream scope, these do not undermine the main contribution. Overall, I believe this paper makes a solid empirical and methodological contribution.